# A Yellow Fever 17D Virus Replicon-Based Vaccine Platform for Emerging Coronaviruses

**DOI:** 10.3390/vaccines9121492

**Published:** 2021-12-16

**Authors:** Nadia Oreshkova, Sebenzile K. Myeni, Niraj Mishra, Irina C. Albulescu, Tim J. Dalebout, Eric J. Snijder, Peter J. Bredenbeek, Kai Dallmeier, Marjolein Kikkert

**Affiliations:** 1Center of Infectious Diseases LU-CID, Department of Medical Microbiology, Leiden University Medical Center, Albinusdreef 2, 2333 ZA Leiden, The Netherlands; nadia.oreshkova@wur.nl (N.O.); S.K.Myeni@lumc.nl (S.K.M.); i.c.albulescu@uu.nl (I.C.A.); T.J.Dalebout@lumc.nl (T.J.D.); E.J.Snijder@lumc.nl (E.J.S.); P.J.Bredenbeek@lumc.nl (P.J.B.); 2Laboratory of Virology and Chemotherapy, Molecular Vaccinology and Vaccine Discovery, Department of Microbiology, Immunology and Transplantation, Rega Institute, KU Leuven, Herestraat 49 Box 1043, 3000 Leuven, Belgium; Niraj_Mishra@intaspharma.com (N.M.); kai.dallmeier@kuleuven.be (K.D.)

**Keywords:** coronavirus, vaccine platform, YFV-17D, YF17D, MERS-CoV, SARS-CoV, SARS-CoV-2, S1, RBD, spike protein, neutralizing antibodies

## Abstract

The tremendous global impact of the current SARS-CoV-2 pandemic, as well as other current and recent outbreaks of (re)emerging viruses, emphasize the need for fast-track development of effective vaccines. Yellow fever virus 17D (YF17D) is a live-attenuated virus vaccine with an impressive efficacy record in humans, and therefore, it is a very attractive platform for the development of novel chimeric vaccines against various pathogens. In the present study, we generated a YF17D-based replicon vaccine platform by replacing the prM and E surface proteins of YF17D with antigenic subdomains from the spike (S) proteins of three different betacoronaviruses: MERS-CoV, SARS-CoV and MHV. The prM and E proteins were provided in trans for the packaging of these RNA replicons into single-round infectious particles capable of expressing coronavirus antigens in infected cells. YF17D replicon particles expressing the S1 regions of the MERS-CoV and SARS-CoV spike proteins were immunogenic in mice and elicited (neutralizing) antibody responses against both the YF17D vector and the coronavirus inserts. Thus, YF17D replicon-based vaccines, and their potential DNA- or mRNA-based derivatives, may constitute a promising and particularly safe vaccine platform for current and future emerging coronaviruses.

## 1. Introduction

In the last two decades, the (re)emergence of a series of RNA virus pathogens led to serious disease outbreaks in humans. The most significant, without any doubt, is the severe acute respiratory syndrome coronavirus 2 (SARS-CoV-2) pandemic, with a tremendous, still ongoing global impact since its surge in early 2020. Many research groups and companies around the world are currently developing vaccines against SARS-CoV-2, exploring different designs and testing their safety and efficacy. In the past 1.5 years, after robust preclinical and clinical testing, several vaccines have been approved for public use by the Food and Drug Administration (FDA) and European Medicines Agency (EMA) [1]. Currently, a total of 25 vaccines against SARS-CoV-2 are registered in different countries worldwide [2], which is a huge achievement. Previously, in 2002–2003, SARS-CoV, a close relative of SARS-CoV-2, caused a human epidemic that started in China, but could be efficiently controlled by more-or-less standard containment measures. Later, in 2012, Middle East respiratory syndrome coronavirus (MERS-CoV) was identified in Saudi Arabia; this virus, to date, continues to cause a relatively small number of infections, mostly by direct transmission from dromedary camels and subsequent spread in household or hospital care settings. Other recent epidemics, such as those caused by the Ebola and Zika viruses, are not new to humans, but are causing recurrent outbreaks or spreading to new territories. This recent epidemic and pandemic history, as well as the current pandemic situation, underscore the need for adequate preparedness and fast and efficient production of remedies against (re)emerging infectious diseases.

Yellow fever (YF) is caused by yellow fever virus (YFV), a mosquito-borne pathogen that infects nonhuman primates and humans. YFV is a member of the *Flavivirus* genus in the family *Flaviviridae* and has a single-stranded positive-sense RNA genome of ~11 kb (Figure 1A). The genome encodes a single polyprotein that is cotranslationally cleaved by cellular and viral proteases into the structural proteins (capsid (C), precursor membrane (prM) and envelope (E)) and the nonstructural proteins (NS1, NS2A, NS2B, NS3, NS4A, NS4B and NS5) [3]. Infections in humans are characterized by a combination of symptoms including fever, headache, jaundice, muscle pain, nausea, vomiting and fatigue, which in most cases resolve within 3 to 4 days. However, around 15% of infected individuals develop a severe disease that is characterized by hemorrhagic fever and jaundice and is fatal in 20–50% of the cases [4].

Currently, YF can be controlled by using the yellow fever virus vaccine strain 17D (YF17D), which is considered one of the most effective and safe vaccines developed to date [5]. YF17D is a live-attenuated vaccine that was developed in the late 1930s by Max Theiler through serial passaging of the wild-type YFV strain (Asibi) in embryonated chicken eggs [5]. Since its first use about 80 years ago, more than 800 million doses have been administered to people at risk of being exposed to YFV infection [6]. The vaccine is well tolerated in humans, with highly rare adverse events. A single-dose YF17D vaccination can induce lifelong protective immunity against YFV, even in immuno-compromised individuals [7,8,9].

The remarkable efficacy and safety record of YF17D renders this vaccine particularly attractive for use as a vector platform to develop vaccines against other pathogens. Multiple experimental YF17D vector vaccines were found able to elicit immune responses against heterologous antigens [10]. Three main strategies have been exploited to express such antigens from the YF17D genomic backbone. In one strategy, small amino acid sequences such as known T- or B-cell epitopes were inserted at different positions in the YFV polyprotein, preferably at the junction between the NS2B and NS3 subunits [11,12,13,14], in the *fg* loop of the E protein [14,15,16] and in the NS1 protein [17]. In another strategy, partial or complete sequences of heterologous proteins were inserted at the junction between the E and NS1 proteins [18,19,20,21,22,23,24,25]. Lastly, chimeric viruses have been created by exchanging the YFV glycoproteins with those of other flaviviruses. This latter approach, first described by Chambers and colleagues for Japanese encephalitis virus (JEV) [26], was subsequently developed further into vaccines against JEV and dengue virus (DENV) [27,28] and patented as the ChimeriVax technology for the creation of vaccines against other flaviviruses [29]. Currently, such vaccines have been licensed against JEV (IMOJEV™) and DENV (Dengvaxia^®^), respectively (produced by Sanofi Pasteur). Vaccines based on the same principle have been developed for West Nile virus and for the Zika virus [30], although those vaccines have not been licensed yet.

Despite the strong efficacy and safety record of the YF17D vaccine, rare cases of serious adverse reactions, particularly in neonates and individuals ≥ 60 years of age, have been documented [31,32]. These rare yet serious events, namely yellow fever vaccine-associated viscerotropic (YEL-AVD) and neurotropic diseases (YEL-AND), can be life-threatening and are directly linked to an individual’s failure to control YF17D replication, leading to protracted overshooting viral replication and active viral dissemination. Therefore, an efficacious platform with an even better safety profile will benefit vaccinees otherwise identified as belonging to high-risk groups, such as people above 60 years of age, pregnant and breastfeeding women, infants younger than 6 months and immunocompromised individuals. In the current study, we aimed to develop a YFV-17D-based vaccine platform comprising a replication-competent but propagation-deficient viral (replicon) particles, thereby engineering and additional layer of safety as compared to propagation-competent YFV17D vaccine platforms.

The construction of RNA replicons has been described for different flaviviruses, including Kunjin virus (KUNV), YFV, DENV, JEV, tick-borne encephalitis virus (TBEV), Zika virus ([33,34,35]) and West Nile virus ([36] and reviewed in [37]). Some of them have been used as homologous [33] or heterologous [38,39,40] vaccines in animal studies. The general strategy for creating flavivirus-based RNA replicons comprises an in-frame deletion of the sequences encoding the structural proteins of the flavivirus except for a short sequence at the 5′ end of the capsid protein, which was shown to be essential for genome replication of YFV [41] and KUNV [42].

In our construct, we replaced the prM (except the first 4–6 codons) and E genes of YF17D with a foreign gene sequence, while leaving the capsid protein and the remainder of the YF17D backbone intact (Figure 1A). The few residual prM amino acids at the N-terminus of the foreign antigen (the abovementioned 4–6 codons) facilitate the authentic cleavage between C and prM, resulting in the efficient release of the foreign antigen. As model foreign antigens for the YF-replicon vector platform, we chose the spike (S) proteins of the previously emerged coronaviruses SARS-CoV and MERS-CoV, as well as the well-studied coronavirus model mouse hepatitis virus (MHV). As a common feature, all coronaviruses express spike proteins on the virion surface. The spike protein comprises two subdomains: S1, which mediates host–receptor binding via its receptor binding domain (RBD), and S2, which is responsible for membrane fusion (as exemplified for MERS-CoV in Figure 1B). The S protein, and more specifically the RBD and S1 subdomains, are primary targets for neutralizing antibodies, and therefore, the spike (or these particular domains) is most frequently used for vaccine development against coronaviruses [43,44,45]. Therefore, we aimed to generate replicon particles that express spike subdomains from the three coronaviruses, SARS-CoV, MERS-CoV or MHV. To that end, we transfected cells with the respective YF replicon RNAs while supplementing the YF17D prM and E proteins in trans by cotransfecting the cells with a separate eukaryotic expression plasmid encoding these proteins. We confirmed proper expression of the coronavirus-derived domains by the YFV17D replicons. We also tested the vector flexibility by inserting diverse foreign domains into the replicons. Finally, we show that replicons expressing the S1 subdomains of MERS-CoV and SARS-CoV, when administered to mice, were able to elicit a (neutralizing) antibody response against the respective domains, as well as against YFV.

## 2. Materials and Methods

### 2.1. Cells and Viruses

BHK-21 cells were cultured in Glasgow’s Modified Eagles Medium (Life Technologies, Carlsbad, CA, USA, 21710-025) supplemented with 8% fetal calf serum (FCS), 10% tryptose phosphate broth (Life Technologies, Carlsbad, CA, USA, 18050-039), 10 mM HEPES pH 7.4 (Westburg, Leusden, The Netherlands, LO BE17-737E) and 1% penicillin–streptomycin (Sigma-Aldrich, Saint Louis, MO, USA, P4458). Vero E6 cells were cultured in DMEM (Westburg, Leusden, The Netherlands, BE12-604F) supplemented with 8% FCS and 1% penicillin–streptomycin. Huh7 cells were cultured in DMEM (Westburg, Leusden, The Netherlands, BE12-604F) supplemented with 8% FCS, 1% penicillin–streptomycin and 1% L-glutamine (Sigma-Aldrich, Saint Louis, MO, USA, G7513). All cell lines were grown at 37 °C and 5% CO_2_. The following viruses were used for virus-neutralizing antibody assays: MERS-CoV strains EMC/2012 (NC_019843.3) propagated and titrated in Huh7 cells and SARS-CoV MA15 (DQ497008.1) and Frankfurt-1 (FJ429166.1) propagated and titrated in Vero E6 cells. All viruses except YF17D were handled in BSL3 containment.

### 2.2. Construction of YF17D Replicons with Foreign Inserts

The YF17D replicons (YF-replicons) were constructed into the background of a full-length YF17D cDNA clone in a pACNR vector [46] by standard cloning techniques. All YF17D constructs were placed under the control of the bacteriophage promoter SP6. The foreign protein sequences used, derived from the S proteins of the coronaviruses MERS-CoV, SARS-CoV and MHV, are listed in Table 1.

All foreign sequences were introduced into the YF17D plasmid vector backbone by replacing the sequences encoding the prM and E structural proteins of the YF17D vaccine strain. In order to preserve the authentic cleavage site between the YFV-capsid (C) protein and the prM protein, 4 to 6 of the N-terminal amino acids of prM were retained between C and the foreign insert (Figure 1C). To determine the number of amino acids necessary for cleavage, in silico cleavage predictions were performed for each construct using the SiganlP-4.1 server [47], and the most favorable variant was chosen for each individual domain. In addition, the authentic signal sequence at the N-terminus of S1 (or spike) was omitted when constructing the respective replicons, since the C-terminus of the capsid protein functions as a signal sequence for the downstream protein in the context of the YFV polyprotein (Figure 1C). The furin cleavage site between the S1 and S2 domains was also removed in all constructs in which S1 from the different coronaviruses was used as an insert. All foreign domains were fused at their C-terminus to a trans-membrane region (TMR) of the Sindbis virus (SINV) E2 protein (strain Giessen_2016A, amino acids 693–734 of the polyprotein encoding the structural proteins). Furthermore, the RBD/S1 domains of SARS-CoV and MHV carried two consecutive V5 tags (GKPIPNPLLGLDST) at their N-termini, and the RBD/S1 domains of SARS were also fused to two consecutive HA tags (YPYDVPDYA) at their C-termini, immediately upstream of the TMR. For the construction of the secreted RBD/S1 domains from MERS-CoV spike, the consensus furin cleavage site (RSRR) was inserted at the C-termini of these domains, followed by the six N-terminal amino acids (-SVPGEM-) of S2 of the MERS-CoV S protein. This short sequence was used to provide space between the furin cleavage site and the endoplasmic reticulum membrane. The mCherry gene was inserted in the YF-replicon in a manner similar to that described for the S-derived domains of the three coronaviruses. This sequence was fused to two consecutive V5 tags on its N-terminus and two consecutive HA tags on its C-terminus to facilitate detection via Western blotting.

In addition to the YF-replicon constructs that contained foreign proteins, a delprM/E replicon was constructed, in which the sequence of the YFV-capsid protein TMR was fused to NS1. At the fusion site, the codons of the four N-terminal amino acids of the prM protein (VTLV) were inserted to ensure proper cleavage. This replicon, named “empty” throughout the manuscript, was used as a control replicon. After cloning, the correct sequence of each construct was confirmed by sequencing.

### 2.3. RNA Synthesis and Electroporation

The mMESSAGE mMACHINE™ SP6 Transcription Kit (Invitrogen^tm^, (Thermo Fisher), Waltham, MA, USA) was used for in vitro runoff RNA synthesis of the YF-replicons according to the manufacturer’s instructions, with a few minor modifications. More specifically, the reaction was incubated for 2 h at 42 °C instead of the recommended 37 °C. To prepare the templates for the reactions, plasmid DNA containing the cDNA of the YF-replicons was linearized by digestion with AflII, which cut immediately downstream of the viral 3′-untranslated region (UTR). The DNA template was then purified by phenol–chloroform extraction followed by ethanol precipitation. Subsequently, electroporations (using 1 µg of RNA transcript in 10^6^ BHK-21 cells) were performed using the Amaxa Nucleofector II instrument with kit T and program T20 (Lonza, Bazel, Zwitserland).

### 2.4. Immunofluorescence Assay (IFA)

To visualize the foreign protein domains expressed by YF-replicons, BHK21 cells were electroporated with the respective in vitro transcribed RNAs and seeded on glass cover slips in 12- or 24-well plates and incubated in culture medium at 37 °C and 5% CO_2_. After 16–40 h, cells were fixed with 3% paraformaldehyde (PFA) for 15 min at RT. PFA was removed by aspiration, and cell monolayers were washed three times with phosphate-buffered saline (PBS) supplemented with 10 mM glycine and stored at 4 °C until staining. Further incubation and washing steps were performed at room temperature (RT). For permeabilization, cells were incubated with 0.1% Triton X-100 for 15 min. Blocking was done with 5% FCS in PBS for at least 30 min. Cells were incubated with dilutions of primary and secondary antibodies in PBS for 1 h and washed three times with PBS after each incubation step. The RBD/S1 domains of MERS-CoV were stained with a primary rabbit polyclonal antibody raised against the RBD domain, amino acids 358–588 (kindly provided by Dr. Berend-Jan Bosch, Utrecht University, The Netherlands) and secondary anti-rabbit-Alexa488 (Thermo Fisher, Waltham, MA, USA A-11008) or –Cy3 (GE healthcare, Chicago, IL, USA, PA45011V) antibody conjugates. Endoplasmic reticulum staining was achieved with anti-protein disulfide-isomerase (PDI) monoclonal antibody (ENZO, ADI-SPA-891) and secondary anti-mouse-Alexa Fluor 488 (Thermo Fisher, Waltham, MA, USA, A-11001) or –Cy3 (Jackson ImmunoResearch, West Grove, PA, USA, 715-165-151) antibody conjugates. Nuclei were stained with 1 µg/mL Hoechst 33258. Samples were embedded with Prolong Gold (Thermo Fisher, Waltham, MA, USA,) and imaged using a Zeiss (Oberkochen, Germany) Axioskop 2 or Leica (Wetzlar, Germany) DM6B microscope.

### 2.5. Polyacrylamide Gel Electrophoresis (PAGE) and Western Blotting

Cells transfected with YF-replicons expressing different foreign genes were lysed in Laemmli sample buffer (LSB) (50 mM Tris-HCl 1M pH 6.8, 2% sodium dodecyl sulfate (SDS), 10% Glycerol, 2.5% Mercaptoethanol, 0.02% Bromphenol blue) at 1 day postelectroporation (DPE). In the experiments with secreted forms of the MERS-CoV RBD/S1, cell lysates were harvested at 1 DPE, while supernatants were harvested at 1 or 2 DPE and lysed with 5 × LSB. In some experiments, cell samples were trypsinized, washed and pelleted, and the cell pellets were lysed in 0.5% NP-40 lysis buffer (0.5% NP-40, 50 mM Tris, pH 7.5, 5 mM MgCl_2_) and treated with peptide-N-Glycosidase F (PNGase F, Biolabs, Ipswich, Ma, USA) according to the manufacturers’ instructions. Samples were separated on 10% or 12% SDS–polyacrylamide gels and thereafter blotted on Amersham Hybond P 0.2 PVDF (Polyvinylidene fluoride) membranes (GE Healthcare, Chicago, IL, USA) for Western blot analysis. As a blocking buffer, 1% casein solution or 2.5% bovine serum albumin solution in PBST (PBS supplemented with 0.05% Tween 20) was used. Antibodies were diluted in PBST. Between the blocking and the incubation steps, blots were washed three times with PBST. Membranes were incubated in blocking buffer or diluted antibody solution either for 1 h at RT or overnight at 4 °C. Specific protein bands were visualized on the Uvitec Alliance Q9 Advanced instrument (BioSPX, Abcoude, The Netherlands) using the following antibodies: (1) yellow fever-specific hyperimmune serum from mice (ATCC, discontinued) and secondary anti-mouse-Alexa Fluor^®^ 647 conjugate (Jackson ImmunoResearch, West Grove, PA, US, 716-605-150) or anti-mouse-horseradish peroxidase (HRP)-conjugate (Dako (Agilent), Santa Clara, CA, USA) P0447). In the case in which the HRP conjugate was used, enzyme activity was detected with ECL substrate (Pierce, Appleton, WI, USA, 32132); (2) primary MERS-S1-specific polyclonal rabbit antibody (SinoBiologicals, Beijing, China, 40069-T52-500) and secondary biotinylated polyclonal swine anti-rabbit antibody (Dako (Agilent, Santa Clara, CA, USA), E0353); (3) monoclonal V5 tag-specific primary antibody, clone 2F11F7 (Thermo Fisher, Waltham, MA, USA, 37-7500) and secondary biotinylated anti-mouse antibody (Dako (Agilent), Santa Clara, CA, USA), E0433). For both (2) and (3), anti-biotin–Cy™3 conjugate (Jackson ImmunoResearch, West Grove, PA, US, 200-162-211) was used as a third step; (4) polyclonal goat anti-cyclosporine B (CypB) antibody (Santa Cruz Biotechnology, Dallas, TX, USA, sc-203610) and secondary anti-goat-Alexa Fluor 488 (Thermo Fisher, Waltham, MA, USA, A11055).

### 2.6. Production of Replicon Particles

To produce replicon particles for vaccination of mice, 5 × 10^6^ BHK21 cells were coelectroporated with 5 µg in vitro transcribed RNA of YF-replicons expressing MERS S1, MERS S1 with furin cleavage site or SARS S1 and 5 µg plasmid DNA encoding prM and E of YFV. The cells from each electroporation reaction were seeded in a 25 cm^2^ culture flask and overlaid with growth medium. Supernatants were harvested at 24–28 h postelectroporation, clarified by centrifugation at 4000× *g* for 10 min and kept at −80 °C until use. The titer was determined with the immunoperoxidase monolayer assay (see below) using the 50% tissue culture infectious dose method (TCID_50_). The titers were calculated with the Spearman–Kärber algorithm [48,49]. For the calculations, a well was considered positive if at least one positive cell was detected.

### 2.7. Immunoperoxidase Monolayer Assay (IPMA)

BHK21 cell monolayers were infected with serial dilutions of the replicon particle-containing harvests and fixed at 2 days postinfection with 2% formaldehyde for 1 h at room temperature. The fixative was removed by washing the monolayer 3 times with PBS. For permeabilization, cells were incubated for 5 min with 1% Triton X-100 at RT. Cells were blocked with 5% fetal calf serum diluted in PBST for 30 min at RT and then incubated with primary anti-YFV-NS1 (BioFront Technologies, Tallahassee, FL, USA, BF-087) (time, dilution and temperature) and secondary anti-mouse-HRP conjugate diluted PBST for 1 h at 37 °C. Between the incubation steps, the cell monolayers were washed 3× with PBST. Peroxidase activity was detected using freshly prepared 3-amino-9-ethyl-carbazole (AEC) substrate (1 mL AEC stock solution (4 mg/mL in DMSO) diluted in 20 mL substrate buffer (0.05 M Na-Acetate pH 5.0), as well as 50 µL 3 % H_2_O_2_). Detection of positive cells was performed by observation under a microscope.

### 2.8. Recombinant Protein Expression

Plasmids encoding the MERS-CoV S1 domain (residues 1–747) and variants fused to the Fc region of the human IgG (MERS-CoV S1-Fc) were kindly provided by Dr. Bart Haagmans (Erasmus MC, Rotterdam, The Netherlands). Similarly, an S1-Fc expression plasmid was constructed for the SARS-CoV domain S1 subunit (residues 1–676) by using standard cloning techniques. The recombinant Fc-fusion proteins were expressed and purified as previously described [50]. Briefly, Fc-fusion proteins were expressed by transiently transfecting HEK-293T cells with the expression plasmids using polyethylenimine (PEI) transfection reagent (Polysciences, Warrington, PA, USA, 23966). Following an incubation time of 7 days in HEK-293T expression medium (293SFM II medium, Life Technologies, Carlsbad, CA, USA), the culture supernatants were harvested and clarified by centrifugation. The proteins were affinity purified using protein A sepharose beads (GE Healthcare, Chicago, IL, USA). Purified proteins were subjected to SDS–PAGE and visualized by Coomassie blue staining and Western blot analysis.

### 2.9. Mouse Experiment

Animals were sourced and experiments performed essentially as described [30,51]. Housing of animals and procedures involving animal experimentation were conducted in accordance with the institutional guidelines approved by the Ethical Committee of the KU Leuven, Belgium under license P140/2016. Animals were housed in individually ventilated type-2 filter top cages in groups of four to five, under controlled conditions of humidity, temperature and light (12 h day/night cycles). Food and water were available ad libitum.

Four randomly assigned groups of 8-week-old IFNα/β/γ R−/− (AG129) male mice were immunized intraperitoneally (i.p.) with 0.4 mL of YF-replicon particles expressing either MERS-CoV S1 (1.2 × 10^5^ TCID50), MERS-CoV S1 soluble/secreted (2.5 × 10^5^ TCID50) or SARS-CoV S1 (0.8 × 10^5^ TCID50) or were mock vaccinated with cell culture medium. Each group consisted of 8 mice except for the mock group, which consisted of 4 mice. A prime-boost vaccination regiment was carried out, and mice were immunized on days 0, 14 and 28 to maximize humoral responses in the mouse model. The animals were weighed at arrival and every 3–4 days afterwards and monitored daily for untoward signs as a result of the vaccination. Sera were collected on days 0, 14 and 28 by jugular vein puncture. All mice were euthanized and terminally bled at day 42 postvaccination (14 days after the last booster).

### 2.10. Indirect ELISA

MERS-CoV, SARS-CoV and YFV-specific antibody titers in mice sera were measured by indirect ELISA using YFV-NS1 (BioRad, Hercules, CA, USA, PI052A), MERS-CoV S1-Fc and SARS-CoV S1-Fc fusion proteins. Briefly, 96-well Nunc Maxisorp plates (Thermo Fisher, Waltham, MA, USA) were precoated by incubation overnight at 4 °C with 100 μL of D-PBS (PBS with Ca^2+^ (0.1 g/L CaCl_2)_/Mg^2+^ (0.1 g/L MgCl_2_·6H_2_O)) containing MERS-CoV S1-Fc or SARS-CoV S1-Fc fusion protein (1 µg/mL) or with carbonate–bicarbonate buffer (pH 9.6) containing YFV-NS1 protein (0.5 µg/mL). Wells were then washed three times with PBST and blocked with blocking buffer (PBS + 0.1% Tween-20, containing 2% BSA) for 30 min at 37 °C. Plates were washed three times in PBST and once in blocking buffer. Next, 100 μL of serially diluted mice sera (in blocking buffer) was added per well, and plates were incubated at 37 °C for 1 h. Plates were washed four times with PBST and then incubated with HRP-conjugated goat anti-mouse IgG (DAKO, (Agilent), Santa Clara, CA, USA), P0447) at 37 °C for 1 h. After four washes, 100 μL of substrate 3,3′,5,5′-tetramethylbenzidine (TMB) (eBioscience (Thermo Fisher, Waltham, MA, USA, Cat no. 00-4201-56) was added per well and incubated for 2–5 min. Reactions were stopped with 100 μL of 12.5% 1 N sulfuric acid (H_2_SO_4_) per well. The absorbance was measured at 450 nm with an EnVision 2105 Multimode Plate Reader (PerkinElmer, Waltham, MA, USA). Mouse anti-YFV-NS1 monoclonal antibody (BioFront Technologies, Tallahassee, FL, USA, BF-087) was used as a positive control for the YFV NS1-based ELISA, while SARS-CoV hyperimmune serum kindly provided by Dr. Bart Haagmans (Erasmus MC, Rotterdam, The Netherlands) was used as a positive control for the SARS S1-based ELISA. Positive antisera against RBD and S1 of MERS-CoV provided by Dr. Berend-Jan Bosch (Utrecht University, The Netherlands) were used as positive controls for MERS-S1-based ELISA. Test sera were diluted by serial fourfold dilutions starting at 1:320 (for MERS-CoV S1 and SARS-CoV S1) and 1:1280 (for YFV NS1), and endpoint titers were expressed as the reciprocal value of the last dilution above a cutoff. The cutoff values were calculated as the average optical density (OD) of all samples from day 0 (preimmune samples) at the lowest dilution plus three standard deviations. For the purpose of graphical representations, samples with undetectable antibody titers were assigned values fourfold lower than those of the starting dilutions, which corresponded to the nearest dilution that could not be measured (80 MERS-CoV S1 and SARS-CoV S1 and 320 for YFV NS1). All serum samples were tested in duplicates. Data were represented as geometric mean end titers.

### 2.11. Virus Titration

The tissue culture infective dose 50 (TCID_50_) endpoint dilution method was used for virus quantification. Titers were calculated using the Spearman–Kärber algorithm. MERS-CoV and SARS-CoV were titrated on Huh7 cells and Vero E6 cells, respectively. Cells were seeded at a density of 10,000 cells/well in 96-well clusters and infected on the following day with virus stocks serially diluted in EMEM/2%FCS/PS/L-Glut media. After an incubation period of 3 days at 37 °C and 5% CO_2_, the cells were fixed with 37% formaldehyde (final concentration of 7.4%) for a minimum of 8 h and then stained with a crystal violet solution for 5–10 min. The excess color was removed by repeated washing with tap water, and the plates were left to dry before assessing the presence or absence of an intact cell monolayer.

### 2.12. Virus Neutralization Test (VNT)

Test sera were heat-inactivated at 56 °C for 30 min, and serial 2-fold serial dilutions were prepared starting from a 1:20 dilution in EMEM/2% FCS/PS/L-glut medium. Each serum dilution was prepared in duplicate (from 2 independent dilutions) and was preincubated (1:1 volumes) with approximately 100 TCID_50_/75 μL MERS- or SARS-CoV for 1 h at 37 °C. Then, serum–virus mixtures were added to monolayers of Huh7 or Vero E6 cells, respectively, seeded at a density of 10,000 cells/well in 96 wells plates the day before. After an incubation period of 3 days at 37 °C/5% CO_2_, the cell monolayers were fixed with a final concentration of 7.4% formaldehyde and stained with crystal violet as described above. A back titration was performed with each experiment to verify the titers of the viruses used in each experiment. The titers ranged between 26 and 262 TCID_50_/well. The reproducibility of the VNTs was affirmed by including the same positive control sera in each run. Titers were expressed as the reciprocal value of the last dilution that completely inhibited the virus-induced cytopathogenic effect. The titers were determined in each of the duplicates, and a mean titer was calculated. For the purpose of graphical representation, samples with undetectable antibody titers were assigned values twofold lower than the lowest detectable titer (titer 10), which corresponded to the nearest dilution that could not be measured (titer 5).

### 2.13. Statistical Analysis

ELISA titers against YFV-NS1 (were compared with nonparametric ANOVA (Kruskal–Wallis test), because the data were not normally distributed. ELISA titers against MERS-CoV S1 and neutralizing titers against MERS-CoV and SARS-CoV were compared with a *t*-test after establishing the data’s normality with a D’Agostino–Pearson test. Data at 42 days postvaccination were compared. All titers were log-transformed before plotting and statistical analysis. The analysis was performed with GraphPad (San Diego, CA, USA) Prism software, ver. 8.3.0.

## 3. Results

### 3.1. Design and Characterization of YF-Based RNA Replicons Expressing Partial or Full-Length MERS-CoV S Protein

For the generation of a YF17D-replicon vaccine platform, the viral gene sequences encoding the prM and E genes of YFV were replaced with a gene sequence encoding (part of) a foreign viral protein of interest (Figure 1A,B). When designing such YF-replicons, first, it was important to preserve the authentic membrane topology of the YFV proteins in order not to disrupt their proteolytic maturation and to obtain viable replicons. Second, we aimed to achieve expression of the foreign protein on the cellular surface, assuming that it could increase antigen exposure to immune cells, thereby augmenting the immune response. For preservation of the membrane topology, it is pertinent that in the natural YFV polyprotein, the sequence encoding the transmembrane anchor (TMA) of the capsid protein serves as a signal sequence for the prM protein, while the TMA of the E protein serves as a signal sequence for the NS1 protein (Figure 1C, left panel). Because prM and E were replaced by a foreign protein in the polyprotein expressed by the YF-replicons, the inserted sequence was preceded by the TMA sequence of the capsid protein to ensure the intended membrane topology and proteolytic processing in the lumen of the endoplasmic reticulum (ER). For membrane anchoring of the foreign protein and correct expression of the downstream NS1 protein, the TMA of the E protein could be used. However, previous studies have shown that this TMA has an ER retention signal, which is consistent with virus assembly occurring at ER membranes [52]. Therefore, using the E-TMA would likely results in the accumulation of the foreign protein mainly in the ER instead of on the cell surface, which contradicts the second requirement for successful replicon design, as outlined above. To achieve expression on the cell surface, a TMA from the E2 glycoprotein of Sindbis virus (SINV) was used instead of the E-TMA (Figure 1C, right panel). Unlike YFV, which assembles at the ER, SINV assembles at the plasma membrane, and the envelope glycoprotein E2 is efficiently exported to the cell surface [53]. We therefore assumed that a foreign antigen carrying the SINV E2 TMA would also reach the cell surface.

To investigate the expression and processing of the YF-replicon proteins, as well as the inserted foreign proteins, BHK-21 cells were transfected with in vitro transcribed RNA of replicons designed to express the RBD or S1 regions of the MERS-CoV S protein. Full-length RNA of the YF17D RNA was used as a control. Cell lysates were analyzed by Western blotting, and bands corresponding to NS1 and NS4B were readily detected in the YF17D and the replicon-containing lanes, revealing expression and correct cleavage of these YF proteins (Figure 1D, lanes 2–4). The E protein was detected only in the samples with full-length YF17D, as expected (Figure 1D, lane 2), since the replicons lacked the E gene. Next, the expression of the MERS-CoV derived RBD and S1 domains was evaluated in cell lysates after transfection with in vitro transcribed RNA of RBD or S1-expressing replicons, respectively. To control for specificity of RBD/S1 protein detection, an empty replicon was used, and its replication was confirmed by immunofluorescence microscopy and Western blotting (data not shown). Both the RBD and S1 domains were detectable on the Western blot in the respective samples (Figure 1E, lanes 1 and 3), while the control sample containing the empty replicon did not yield any signal (Figure 1E, lanes 5, 6). Since S1 is heavily glycosylated and the smaller RBD domain has two predicted N-glycosylation sites [54], we also confirmed the N-glycosylation status of the RBD and S1 domains as expressed by YF-replicons. Cell lysates were treated with Peptide:N-glycosidase F (PNGase), which removes N-linked glycans from glycoproteins. The PNGase treatment enhanced the electrophoretic mobility of both products, as visualized on Western blots (Figure 1E, compare lane 1 to 2, and lane 3 to 4). The RBD and S1 products migrated with estimated molecular masses of ~32 kDa and ~87 kDa, respectively, which matched the calculated sizes of the nonglycosylated versions of these domains, revealing that both were indeed glycosylated when expressed from our YF17D replicons in BHK-21 cells.

Finally, we investigated whether a protein as large and complex as the full-length MERS-CoV S protein (1353 amino acids) could be incorporated and expressed by a YF-replicon. For that purpose, a full-length S protein gene was introduced in the replicon in a manner similar to that used for the smaller RBD and S1 domains. Cells were electroporated with in vitro transcribed replicon RNA and probed for expression of the S protein by Western blot analysis. The full-length S protein was detectable in cell lysates, and PNGase treatment again induced a mobility shift indicative of N-linked glycosylation of the product, as expected (Figure 1F).

### 3.2. Cellular Localization of the Foreign RBD and S1 Expression Products

As explained above, the coronavirus S domains were fused to the TMA of SINV E2 protein to ensure their transport to the cell surface. To confirm the cell surface localization of the MERS-CoV RBD and S1 domains, cells containing the respective YF-replicons were either permeabilized or not before being immunolabeled for RBD/S1 and analyzed by fluorescence microscopy. As a control for membrane permeabilization, cells were costained with an antibody recognizing the cellular protein disulfide-isomerase (PDI), which is routinely used as a marker for the ER lumen. All permeabilized cells were strongly labeled for PDI, while nonpermeabilized cells remained unstained. The MERS-CoV RBD and S1 products were detected in both permeabilized and nonpermeabilized cells, confirming that they were both expressed and in part transported to the cell surface (Figure 2A).

To demonstrate that the TMR of SINV E2 was essential for surface expression of the foreign insert in the YF-replicon, replicons were built that expressed the RBD domain with the authentic TMR of the YFV E protein. The surface expression of the RBD domain with either TMR was compared in permeabilized and nonpermeabilized cells. While in both cases, the RBD domains were abundantly labeled in permeabilized cells, a distinct staining of the surface of nonpermeabilized cells was observed only for the RBD carrying the SINV E2 TMR, confirming that this TMR was the key to successful surface expression (Figure 2B).

### 3.3. The YF-Replicons Can Express Diverse Inserts in Either Cell-Associated or Secreted Form

A highly important feature for a vector platform is its capability to incorporate and express diverse foreign inserts. Additionally, the immune response can be fine-tuned by presenting the foreign antigens in either cell-associated or secreted form. These features would potentially equip the platform for use against different pathogens. As a proof of concept for secreted antigen production, we aimed to generate secreted version of the MERS-CoV RBD and S1 domains in addition to products that were anchored to the plasma membrane. To that end, a consensus furin cleavage site [55] was introduced at the C-terminus of both domains, immediately upstream of the TMR (Figure 3A,E). Furin is a ubiquitous proprotein convertase that cleaves precursor proteins along the cell’s secretory pathway [56]. Therefore, proteins with a furin cleavage site that is located in the ER lumen are subjected to furin cleavage. To test for the secretion of the two MERS-CoV S domains, cells were electroporated with in vitro transcribed YF-replicon RNA expressing the RBD or S1 domain with or without an added furin cleavage site. Lysates from cell fractions and supernatants were harvested at day 1 or 2 posttransfection and probed for the presence of the RBD or S1 domains. The TMR-anchored RBD was found exclusively in the cellular fraction, while the RBD from the construct containing the furin cleavage site was also detected in the supernatant (Figure 3B). Unexpectedly, the S1 domain was found to be secreted in the case of both the TMR-anchored protein and the protein containing the engineered furin cleavage site. However, furin cleavage did enhance S1 secretion, which was clearly visible when samples taken at day 1 posttransfection were compared (Figure 3C, lanes 5 and 8). There was a natural furin cleavage site (PRSVR↓S) between the S1 and S2 domains of the MERS-CoV S protein, but that site was removed in the construct expressing the TMA-anchored S1 (Figure 3D), and an in silico prediction (ProP 1.0 server online) suggested its successful elimination. Currently, the mechanism of cleavage of the TMR-anchored form of S1, resulting in its (partial) secretion, is unclear.

To further elucidate the capacity of the YF-replicon vector to incorporate and express foreign proteins, we constructed replicons expressing the RBD and S1 domains from two other coronaviruses—MHV and SARS-CoV (Figure 4A). To facilitate detection of the expressed proteins, two consecutive V5 tags were fused to the N-terminus of each insert. A replicon expressing a reporter gene (mCherry) was also constructed. The mCherry protein was tagged in an identical manner as the MHV and SARS-CoV domains, and its expression was confirmed using a fluorescence microscope (data not shown). All heterologous expression products were detected in lysates from cells containing the respective YF-replicons, as presented in Figure 4B. Notably, both MHV RBD and S1 were expressed at much lower levels than SARS-CoV RBD and S1 and mCherry. The expression of the MHV-derived domains was consistently low in multiple experiments, even in sample lysates normalized for the percentage of replicon-positive cells. Lastly, the N-linked glycosylation status of the RBD and S1 domains of SARS-CoV was investigated by treating transfected cell lysates with PNGase. This resulted in the generation of discrete, faster-migrating bands upon SDS–PAGE as compared to those of the nontreated samples (Figure 4C). The PNGase treatment did not influence the migration of the mCherry bands, which was expected (NetNGly 1.0 server [57]).

### 3.4. YF-Replicons Are Packaged into Infectious Particles and Express the Encoded Foreign Protein Inserts

For further development of the vaccine platform, it was essential to equip the YF-replicon RNA with a suitable packaging vehicle to facilitate its delivery to recipient cells. To this end, we utilized an expression plasmid encoding the YFV structural proteins prM and E, provided in trans to cells that contained YF-replicons by cotransfection, thereby enabling the formation of infectious particles containing the YF-replicon RNA. To validate particle production and infectivity, as well as expression of the foreign protein of interest following infection, BHK-21 cells were coelectroporated with in vitro transcribed YF-replicon RNA expressing the RBD of MERS-CoV and a plasmid expressing the YF17D prM and E proteins. Control electroporations were performed with either the replicon RNA or the prM and E-expressing plasmid alone. At 24 and 48 h postelectroporation, supernatants were harvested from these transfected cells and incubated with untreated BHK-21 cells for 24 h. Following that period, expression of MERS-CoV spike RBD was investigated. RBD expression was detected only in cells incubated with supernatants harvested from coelectroporated cells, confirming the presence of infectious particles in those supernatants (Figure 5). As expected, no RBD-positive cells were detected in cells incubated with control supernatants harvested from the replicon RNA or the prM/E plasmid single electroporations. This shows that the replicon alone (or the expression plasmid encoding prM/E alone) could not produce infectious particles, indirectly confirming the single-cycle capacity of the replicon particles and thereby the safety of the replicon vaccine platform. Similar experiments were performed with replicons expressing S1 of MERS-CoV, S1 of MERS-CoV with a furin cleavage site, S1 of SARS-CoV, S1 of MHV and mCherry. The infectivity titers of the replicon particle harvests ranged between 10^6.5^ and 10^7.1^ TCID_50_/mL for all constructs except MHV S1, which consistently had titers of approximately a log10 lower.

### 3.5. S Protein-Expressing YF-Replicons Induce Neutralizing Antibodies against Both MERS-CoV and SARS-CoV in Mice

Finally, the immunogenicity of YF-replicons expressing the different betacoronavirus antigens was evaluated in vivo in AG129 mice. These mice are deficient in receptors for type I and II interferons, rendering them very permissive to YF17D. Their adaptive immune system, however, is intact, and they can mount B- and T-cell directed immune responses upon immunization. Hence, they are widely used as an infection model for YF17D [58,59,60] and for the assessment of the safety [25] and efficacy of YF17D-based vaccines [30,51,61]. Mice were vaccinated with YF-replicon particles (using repeated dosing of about 10^5^ infectious particles per animal and injection) expressing one of the following inserts: MERS-CoV S1, MERS-CoV S1 with a furin cleavage site (MERS-CoV S1secr) or SARS-CoV S1 (Figure 6A). The respective antibody responses induced by this vaccination were measured. A group of mice vaccinated with culture medium served as a mock-vaccinated control. Responses against the vector were evaluated by measuring the anti-YFV NS1 protein antibodies using ELISA. Anti-NS1 total binding antibodies were detectable already at day 14 in 7/8 of the mice vaccinated with the MERS-CoV S1-replicon, in 4/8 of the mice vaccinated with the MERS-CoV S1secr-replicon and in 2/8 of the mice vaccinated with the SARS-CoV S1-replicon. After the first booster, anti-NS1 antibody ELISA titers increased in all vaccination groups, and all but one mouse from the SARS-CoV S1-replicon-vaccinated group had detectable levels of antibodies. Two weeks after the second booster (at day 42 post-prime vaccination), all mice had seroconverted for YFV NS1 antibodies, and there were no statistical differences between the geometric mean titers (GMT) of the three groups, regardless of the vaccine used (mean titers for MERS-CoV S1-replicons: 10^4.4^, MERS-CoV S1secr-replicons: 10^4.6^ and SARS-CoV S1-replicons: 10^4.5^) (Figure 6B–D). None of the mock-vaccinated mice had detectable anti-NS1 antibodies (data not shown).

Antibody responses against the coronavirus S antigen expressed by the replicons were evaluated by both ELISA and virus neutralization assay. Total binding antibodies against MERS-CoV S1 were already detectable at day 14 by ELISA in 7/8 and in 6/8 of the mice vaccinated with MERS-CoV S1-replicons and MERS-CoV S1secr-replicons, respectively (Figure 6E,F). Similarly to the response against YFV NS1, booster vaccinations resulted in an increase in total binding antibody titers against MERS-CoV S1. On day 42, all mice had measurable antibodies against S1, and there was no statistical difference in GMT between the two groups (mean titers for MERS-CoV S1-replicons: 10^5.7^ and MERS-CoV S1secr-replicons: 10^5.9^). Induction of neutralizing antibodies in these groups followed a pattern similar to that of the total binding antibodies detected with ELISA (Figure 6H,I). Fourteen days after the prime vaccination, 7/8 and in 6/8 of the mice vaccinated with MERS-CoV S1-replicons and MERS-CoV S1secr-replicons, respectively, had detectable neutralizing antibodies. Titers increased as a result of the following booster vaccinations. Finally, and most importantly, on day 42, all vaccinated mice had detectable levels of neutralizing antibodies without any significant statistical difference in GMT between the groups (mean titers for MERS-CoV S1-replicons: 10^2.5^ and MERS-CoV S1secr-replicons: 10^2.5^).

In mice vaccinated with SARS-CoV S1-replicons, antibodies were detectable by ELISA in 5/8 of the animals 14 days after the prime vaccination (Figure 6G). Titers of the antibodies increased following the booster vaccinations, and on day 42, all the mice had seroconverted. A virus neutralization test performed with SARS-CoV strain MA15, which is the parental strain for the S1 protein expressed by the YF-replicon, revealed detectable antibodies in only 1 out of 8 mice 14 days after prime vaccination. The two booster vaccinations resulted in a gradual increase in the titers and detectable neutralizing antibodies in all the mice on day 42 (Figure 6J). Interestingly, when a virus neutralization assay was performed with a different strain, Fr-1, antibodies were not detectable in any of the mice on day 14, and after the two booster vaccinations, they were detectable in only 4 of the 8 mice. On day 42, there was a significant difference in neutralizing titers against the strains MA15 and Fr-1 (Figure 6J).

No antibodies were detected against SARS-CoV S1 or MERS-CoV S1 in any of the mock-vaccinated mice (data not shown). During the whole experiment, no mouse showed any sign of visible disease, and intergroup weight changes remained comparable (results not shown).

## 4. Discussion

The current SARS-CoV-2 pandemic is showcasing the importance that novel platforms can have for the fast development of antiviral vaccines against emerging viruses that threaten global populations, and in particular, the need for versatile platforms from which safe and effective vaccines can readily be derived. The undeniable success of the novel mRNA-based vaccines produced by Biontec/Pfizer and Moderna [62] has opened important perspectives for innovative vaccine approaches in general. The adenovirus-based vaccine platforms used by AstraZeneca and Janssen Vaccines and Therapeutics and their academic partners are also considered safe and efficacious by the European Medicines Agency [63,64]. However, the often high reactogenicity and in particular the rare yet considerably severe side effects of these vaccines, such as thrombosis [65,66] and myocarditis [67], have drawn attention now that they are being used in large populations. Hence, there is still a need to explore alternative, safe and flexible vaccine platforms for the development of second-generation betacoronavirus vaccines.

YF17D is one of the most successful human vaccines to date. Its remarkable efficacy has been attributed to the activation of distinct subsets of dendritic cells through engagement of multiple toll-like receptors (TLR 2, 7, 8, 9) [68], signaling via RIG-I and MDA-5 [69] and stimulation of multiple innate immunity pathways in infected cells [70,71]. As a result, a broad and long-lasting adaptive immune response is induced. Single-dose vaccination with YF17D elicits antibody responses in >90% of the vaccinees; these responses persist as long as 40 years in at least 80% of the vaccinated individuals (reviewed by Gotuzzo et al. [7] and Monath and Vasconcelos [72]). Cellular responses are characterized by a mixed Th1/Th2 signature [73], accompanied with robust, polyfunctional and long-lasting CD8 T-cell responses [74,75]. The outlined immunogenicity of YF17D is combined with an excellent safety record, although severe neurotropic (YEL-AND) and viscerotropic disease (YEL-AVD) have on rare occasions been recorded after YF17D vaccination, resembling conditions that are also associated with the yellow fever virus infection itself [76,77,78]. This exceptional vaccine immunogenicity and efficacy combined with a favorable safety profile explains the interest for using YF17D vaccine as a broad platform for vaccines against other pathogens.

In this work, we demonstrated the design and characterization of YF17D-based replicons as a vector platform for emerging coronaviruses. The advantage of using replicons instead of propagation-competent viruses is an added layer of safety. The highly safe profile of the replicons becomes clear in the context of the immune-suppressed AG129 mouse model used in our study to test immunogenicity. These mice lack receptors for both type I (α/β) and type II (γ) interferons and develop a transient viscerotropic and lethal neurotropic disease when infected with even as little as 1 PFU of YF17D [58,60]. In the current study, mice were vaccinated with replicons three times in two-week intervals, and we did not observe any adverse effects in these mice, confirming the marked safety of the YF-replicon vaccine platform even in this immune compromised model. This platform would therefore probably be beneficial to use in high-risk population groups for which the spreading-competent YF17D is not recommended, such as infants younger than 9 months, pregnant women, people older than 60 years and immunocompromised individuals [79]. A recent report described efforts of sequence adaptation of the yellow fever virus backbone to diminish adverse effects causing viscerotropic and neurological disease [80] as another way to increase safety of the vaccine. However, the intrinsic risks of reversion of these mutations in the full live virus is still present, which is not the case for the replicon approach described here.

Using the platform as a vehicle for other viral antigens raises the question of whether preexisting YFV immunity, as may exist in parts of the target population, might interfere with the efficacy of such a vaccine platform. Investigations of this possible negative effect suggest that for different viral vector platforms such as Ad (adenovirus) and live-attenuated viral vectors such as alphavirus, MeV (measles virus) [81]) or HSV (herpes simplex virus), the effect depends on many factors but does not necessarily diminish the efficacy of these vaccines [82,83]. In the case of the YF-replicon vaccine, while the replicon particles did carry prM and E proteins, they did not further express these structural YF17D proteins, of which the E protein is thought to play a major role in eliciting YFV neutralizing antibodies [84,85]. A replicon vaccine may therefore trigger less neutralizing response to the YFV itself, leaving room for effective repeated vaccinations. The potential problem of preexisting immunity against YFV for the efficacy of the replicon-vaccine platform should be further investigated, especially in areas where yellow fever is endemic. Nevertheless, in case anti-YF neutralizing antibodies would finally hamper the efficacy by which the replicons are transduced in vivo, the prM/E helper proteins can readily be exchanged during particle production for those of antigenically distinct flaviviruses belonging to another less prevalent serogroup than YFV. Additionally, a small dose of imlifidase (an enzyme that breaks down IgG antibodies) could be supplemented with YF-replicons for transient suppression of YF-neutralizing antibodies [86].

Studies exploring correlates of protection against MERS-CoV, SARS-CoV and SARS-CoV-2 unambiguously show that neutralizing antibodies have a key role in protection against betacoronaviruses [87,88,89,90]. Therefore, our main goal in the present study was to evaluate the (neutralizing) antibody responses after vaccination. Induction of antibodies against YF17D in the context of a vaccine vector has been reported in studies performed both in mice [14,19,21,23,25] and in rhesus macaques [16,20,25]. Macaques, similarly to humans, are highly permissive for YF17D infection, and as a consequence, single-shot vaccination is sufficient to elicit antivector antibodies in this species. In immunocompetent mice, however, YF17D is severely attenuated [59,91]. Therefore, although robust antibody and cellular responses have been reported in wild-type mice following YF17D vaccination [92,93], using this vaccine as a vector usually requires a booster vaccination regimen in such mice [14,19,21,23]. In this regard, it was plausible that the replicon-based vaccines as investigated here, which do not spread within the host, would be even less potent for induction of antibody responses in mice. We therefore opted for using the AG129 mice, which are known to be very susceptible to infection with YF17D, and a fairly aggressive and short booster regimen that may not reflect any relevant clinical schedule. In these mice, replicon immunization resulted in induction of antibody responses against the foreign antigen of interest, as well as YFV-NS1. Detection of both insert-specific and anti-NS1 antibodies implies that the YF-replicon RNA actively amplified intracellularly in vivo and that the viral nonstructural proteins, as well as the foreign viral proteins, were expressed to induce immune responses. Induction of insert-specific antibodies was also observed in mice vaccinated with spreading-competent viral vectors [12,19,21,23,25]. To our knowledge, our study is the first to show induction of insert-specific antibodies elicited by a nonspreading YF-replicon in a mouse model.

Overall, the responses against the YFV-NS1 protein developed at later timepoints than antibodies against S1 (both for MERS-CoV and SARS-CoV). This discrepancy could result from differences in immunogenicity and/or expression levels of NS1 and S1, surface exposure (we designed the S1 to be surface expressed, while NS1 was not expressed on the cell surface), or simply differences in the ELISA sensitivity, for example, due to possible different affinities of the antibodies used. Importantly, binding antibodies against the foreign insert were detectable in almost all mice already after two vaccinations and in all mice after three vaccinations, which shows that the vaccine was immunogenic and produced an effective response for the antigen of interest.

In our study, we included YF-replicons expressing two different forms of the S1 protein—a cell-surface-expressed, membrane-bound S1 and a secreted S1. It has been previously demonstrated that the cellular localization of the expressed antigen may play a role in both the magnitude and the quality of the antibody response in mice [94,95,96,97,98], although the findings are controversial and suggest that each antigen and administration route should be investigated for efficacy separately. We did not observe significant differences in the antibody titers between the groups of mice vaccinated with MERS-CoV S1-replicon and MERS-CoV S1 secreted-replicon. However, our vaccine vector characterization revealed that part of the membrane-bound S1 was also detectable in the supernatant of replicon-expressing cells (Figure 3), which may be the reason why no clear differences between the two groups of mice were observed. The release of the membrane-bound form of the S1 could be explained with the presence of a cryptic cleavage site in S1, which may cause part of S1 to be released in the cytosol. Additional investigation could identify a cryptic cleavage site in S1 and would allow the evaluation of the differences between the two forms of antigen (secreted or not) in terms of immunogenicity in the context of the YF-replicon vector in future studies.

The antibody responses against SARS-CoV S1 seemed lower than those against the S1 of MERS-CoV. However, these results are hard to compare, since the ELISA and the VNT used for SARS-CoV and MERS-CoV were performed with different proteins and viruses, respectively. Therefore, the observed differences might originate in different test sensitivities. Striking, however, was the difference in neutralizing antibody levels against the two strains of SARS-CoV, MA15 and Fr-1. MA15 is a mouse-adapted strain of SARS-CoV and differs by six amino acids from isolates from humans [99], including Fr-1. One of these differences is a H436Y substitution in the RBD of MA15; Fr-1 carries the original amino acid. It has been reported that mice vaccinated with wild-type SARS-CoV (strain Urbani) can survive an otherwise lethal challenge with MA15. However, in that model, the animals did show morbidity, characterized by ~10% weight loss, suggesting that the MA15 vaccine was not completely protective against the challenge with the heterologous strain [99]. These data and our findings collectively suggest that if a YF-replicon vaccine were designed to protect humans, and therefore encoded an S1 sequence derived from a human virus isolate, the vaccine efficacy might be underestimated in the setting of an MA15 challenge model.

With the YF-replicon vector described in this study, we were able to express different inserts, including the RBD and S1 domains of the spike protein of two betacoronaviruses, MERS-CoV and SARS-CoV, the full-length MERS-CoV spike protein and the reporter protein mCherry. Expression of the RBD and S1 domains of the spike protein of MHV, another betacoronavirus, was also achieved. However, the expression levels of MHV spike-derived domains were consistently lower than those of the other antigens. Since the YF proteins were expressed in similar amounts by all constructs, as observed by flow cytometry (data not shown) and Western blot analysis (Figure 4B, middle panel; NS4B), we hypothesize that the RBD and S1 domains of MHV are unstable when expressed outside of the context of the full-length MHV S protein and are therefore rapidly degraded. Production of particles containing the MHV S1 replicon consistently yielded titers of at least a *log* lower than when using SARS-CoV S1 and MERS-CoV S1 spike sequences (data not shown). Whether the expressed MHV S domains interfered with the replication of the YF17D backbone and/or expression of the YF17D proteins, or their expression out of the context of the whole spike protein was suboptimal and resulted in fast degradation, remains to be determined. This particular result suggests that, despite the general flexibility of the YF-replicon platform in terms of accommodating different foreign proteins, the successful expression of any foreign insert and the formation of replicon particles will need to be determined experimentally for each individual case.

For a number of the expressed inserts, we demonstrated proper cleavage of the YFV proteins and the inserts and showed the intracellular localization of the foreign inserts and the secretion of constructs with a furin cleavage site. Although the correct folding state of these foreign inserts was not investigated in detail, the fact that neutralizing antibodies were detectable in vaccinated mice is indirect evidence that neutralizing epitopes were exposed to the immune system. We also demonstrated that the protein domains expressed by the replicons had N-linked glycans as expected, which again suggests correct folding. For any future constructs, it remains to be experimentally determined per case whether the protein/domain is presented in a correct way to elicit neutralizing antibodies.

In summary, we here describe the production of YF-replicons that could accommodate different foreign proteins, be packaged into infectious virions and be used as a vaccine that elicits antibodies against the foreign (viral) protein of interest. The replicons had an additional layer of safety as compared to the YFV17D vaccine and did not cause any untoward effect in AG129 mice, while in contrast, infection with YF17D vaccine virus is lethal in this strain. We expect that the replicons will (at least in part) benefit from the exceptional efficacy and immunogenicity of the YF17D backbone (for an opinion on vaccine vectors, see [100]), but that remains to be determined in detail in future studies. The proof of concept, described here, warrants further exploration and characterization of this platform, including analysis of the robustness and durability of the neutralizing antibody responses and cellular immune responses as well as comparison with the YF17D vaccine. Durable protection against infection and transmission of model pathogens in relevant animal models is now to be investigated.

## Figures and Tables

**Figure 1 vaccines-09-01492-f001:**
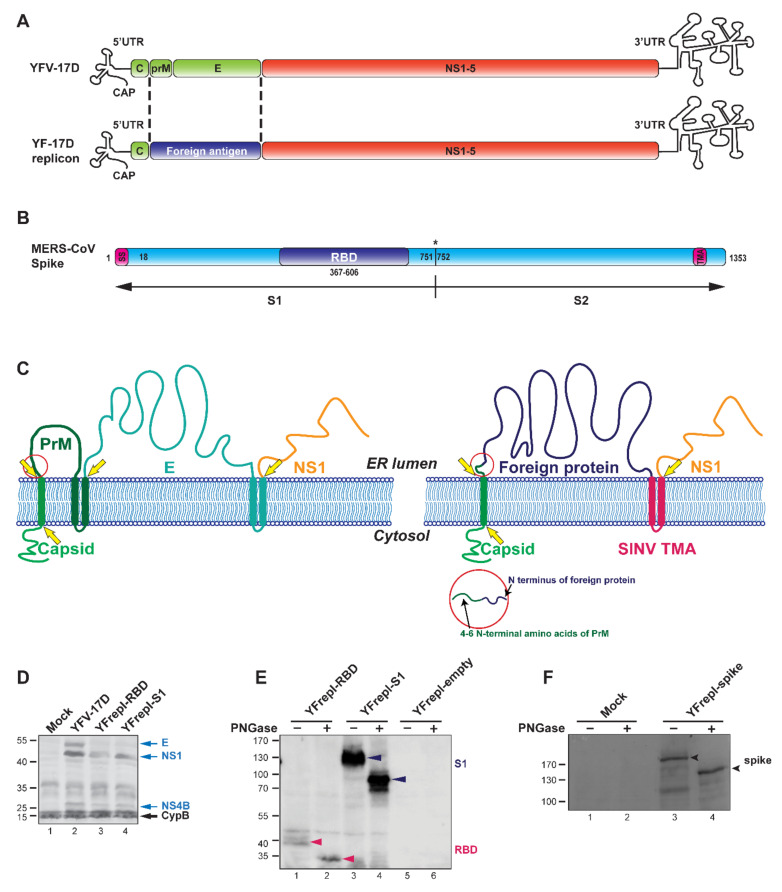
Design and characterization of YF-replicons expressing the RBD or S1 domain of the MERS-CoV S protein. (**A**) Schematic representation of the YFV genome and that of a YF replicon, in which the sequences encoding the glycoproteins prM and E are replaced by a sequence encoding a foreign protein. (**B**) Schematic representation of the MERS-CoV spike protein and its subdomains; SS—signal sequence; TMA—transmembrane anchor. Cleavage site between S1 and S2 is illustrated with an asterisk. (**C**) Cartoon showing the membrane topology of the YFV (left panel) and YF replicon (right panel) polyprotein. The yellow arrows indicate cleavage sites in the viral/replicon polyprotein. The bars that traverse the membrane represent the transmembrane anchors (TMA) of the N-terminally located protein. The red circle highlights the prM amino acid sequence retained at the site of the luminal cleavage that releases the N-terminus of the foreign insert. (**D**) Western blot analysis confirming the expression and processing of YF-replicon polyproteins, as compared to YFV-17D, using a polyclonal serum against YFV. Arrows on the right indicate the viral proteins (blue) and the intracellular chaperon cyclophilin B (CypB), which was used as a loading control (black). (**E**,**F**) Western blot analysis of the expression and glycosylation status of the RBD (red arrow)/S1 (black arrow) domains (**E**) or the full-length S protein (**F**, black arrow), expressed from YF replicons as indicated. Treatment with PNGase revealed the expected glycosylated nature of each of these proteins.

**Figure 2 vaccines-09-01492-f002:**
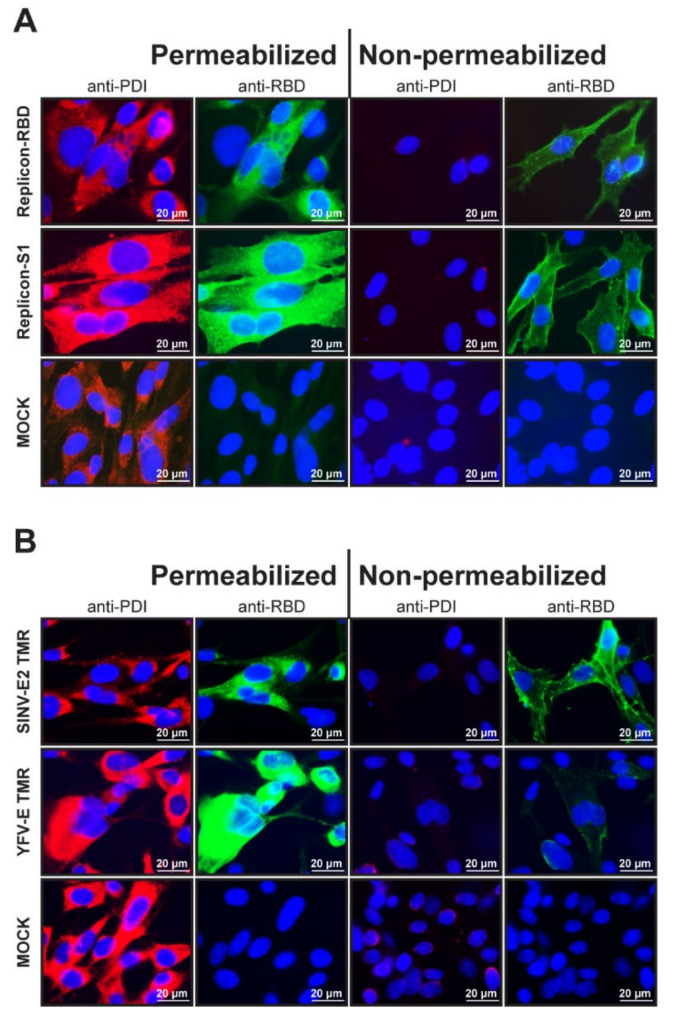
Immunofluorescent images of cells containing YF-replicons that express domains of the MERS-CoV S protein. Cells were either permeabilized with Triton X-100 for intracellular staining or left untreated for surface staining. (**A**) Surface and intracellular expression of the foreign domains RBD or S1. Cells were costained for protein disulfide isomerase (PDI) and MERS-CoV spike RBD. (**B**) Surface and intracellular expression of the RBD domain of MERS-CoV spike protein fused to either the SINV E2 TMR or the authentic YFV-E TMR in cells. Cells were costained for PDI. (**A**,**B**) PDI is stained red and RBD, green, and cell nuclei are visualized with DAPI staining (blue). A 63× objective (with immersion) was used to obtain the images.

**Figure 3 vaccines-09-01492-f003:**
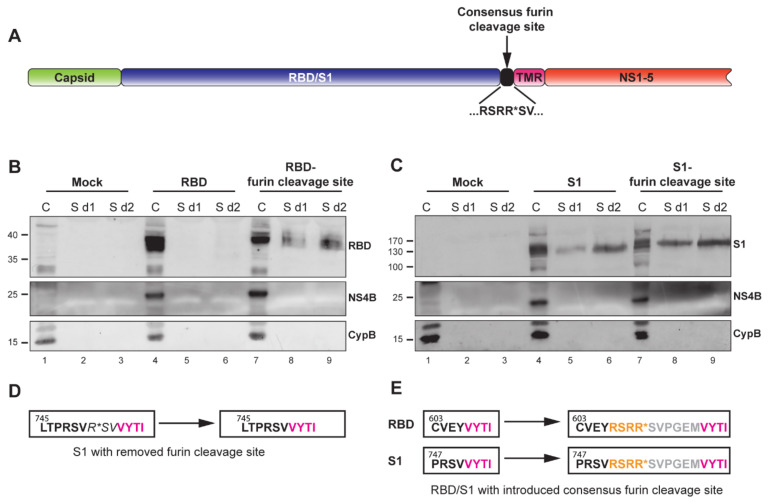
Characterization of MERS-CoV spike RBD- or S1-expressing replicons containing a furin cleavage site at the C-terminus of the insert. (**A**) Schematic representation of the YF-replicons with a consensus furin cleavage site, inserted at the C-terminus of the foreign domain RBD/S1 and upstream of the TMR. Successful furin cleavage results in the release of the RBD/S1 domain from the TMR and secretion in the medium. (**B**,**C**) Western blot analysis of cell lysates and supernatants of cells containing RBD- or S1-YF-replicons with or without engineered furin cleavage sites. Cell lysates were harvested at 1 day postelectroporation and are indicated by the “C” lanes. Supernatants were harvested 1 or 2 days postelectroporation and are indicated as “S d1” and “S d2”, respectively. The quantities of the cell lysates loaded on gel were normalized for the percentage of replicon-positive cells, as determined by flow cytometry (data not shown). The quantity of the supernatants was normalized to correspond to the amount of the loaded cells. The detected proteins are indicated on the right of the Western blot images. NS4B is a yellow fever virus protein and demonstrates the level of expression of the replicons. Intracellular chaperon cyclophilin B (CypB) was used as an internal control. (**D**) Amino acid sequence of the authentic C-terminal part of MERS-CoV S1 with its furin cleavage site (left box) and with the deletion of the furin cleavage site in the YF replicon (right box). (**E**) Amino acid sequence of the C-terminus of the RBD (upper left box, black letters) or S1 domains (lower left box, black letters) and the start of the downstream TMR (left boxes, pink letters) in the YF-replicons without furin cleavage sites and with introduced consensus furin cleavage sites (right boxes, yellow letters). The grey letters show the N-terminal 6 amino acids of the S2 domain of MERS-CoV spike used as a short spacer between the furin cleavage site and the downstream TMR. (**D**,**E**) Numbers illustrate amino acid position in the spike protein.

**Figure 4 vaccines-09-01492-f004:**
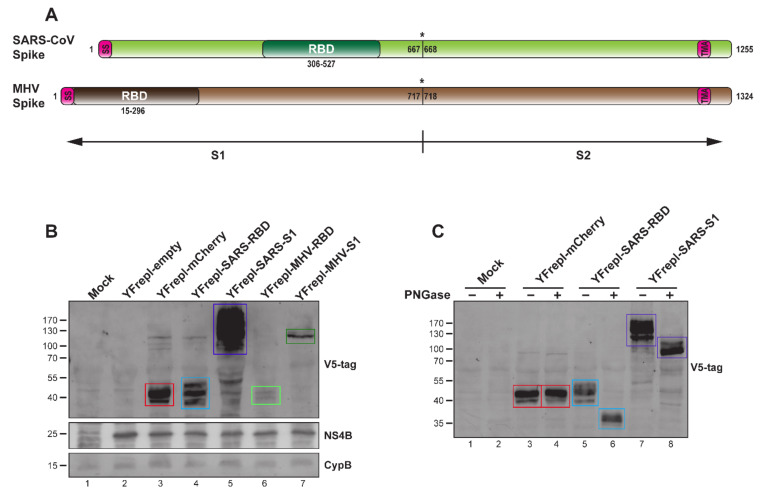
Characterization of YF-replicons expressing various coronavirus inserts. (**A**) Schematic representation of the SARS-CoV and MHV S proteins. SS—signal sequence; TMA—transmembrane anchor. Cleavage site between S1 and S2 is illustrated with an asterisk. (**B**) Western blot analysis of the expression of the RBD or S1 domains of SARS-CoV and MHV, as well as mCherry, in lysates from cells containing the respective replicons. The quantities of the cell lysates loaded on gel were normalized for the percentage of replicon-positive cells, as determined by flow cytometry. (**C**) Western blot analysis of the glycosylation status of the RBD and S1 domains of SARS-CoV or mCherry after PNGase treatment of lysates from cells containing the respective YF-replicons. (**B**,**C**) Bands with the expected size for all inserts are indicated with colored rectangles: red for mCherry, light blue for MERS-CoV RBD, purple for MERS-CoV S1, light green for MHV RBD, and dark green for MHV S1. The antibodies used are shown on the right. A protein size marker is shown on the left. CypB and YFV NS4B were used as internal and replicon expression controls, respectively.

**Figure 5 vaccines-09-01492-f005:**
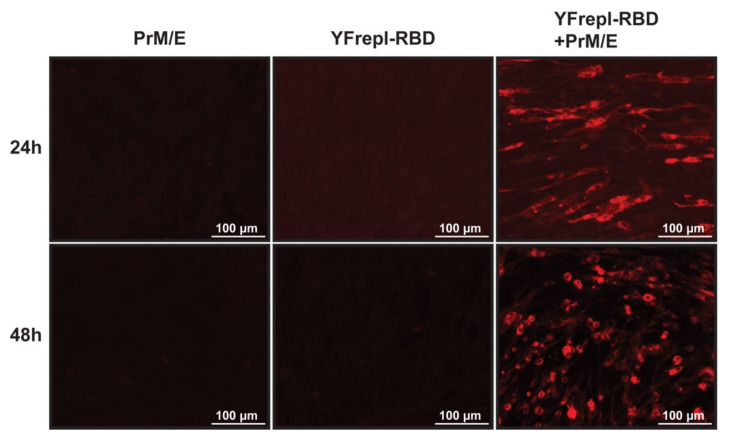
Expression of MERS-CoV RBD, following infection with replicon particles. Supernatants from BHK-21cells that were electroporated with either a plasmid encoding YFV prM/E, in vitro transcribed replicon RNA expressing the MERS-CoV RBD (YFrepl-RBD) or both, were harvested 24 or 48 h postelectroporation and used to infect fresh BHK-21 cells. After 24 h of incubation, the cells were fixated and analyzed by immunofluorescence microscopy. The RBD domain is stained in red. A 20× objective was used to obtain the images.

**Figure 6 vaccines-09-01492-f006:**
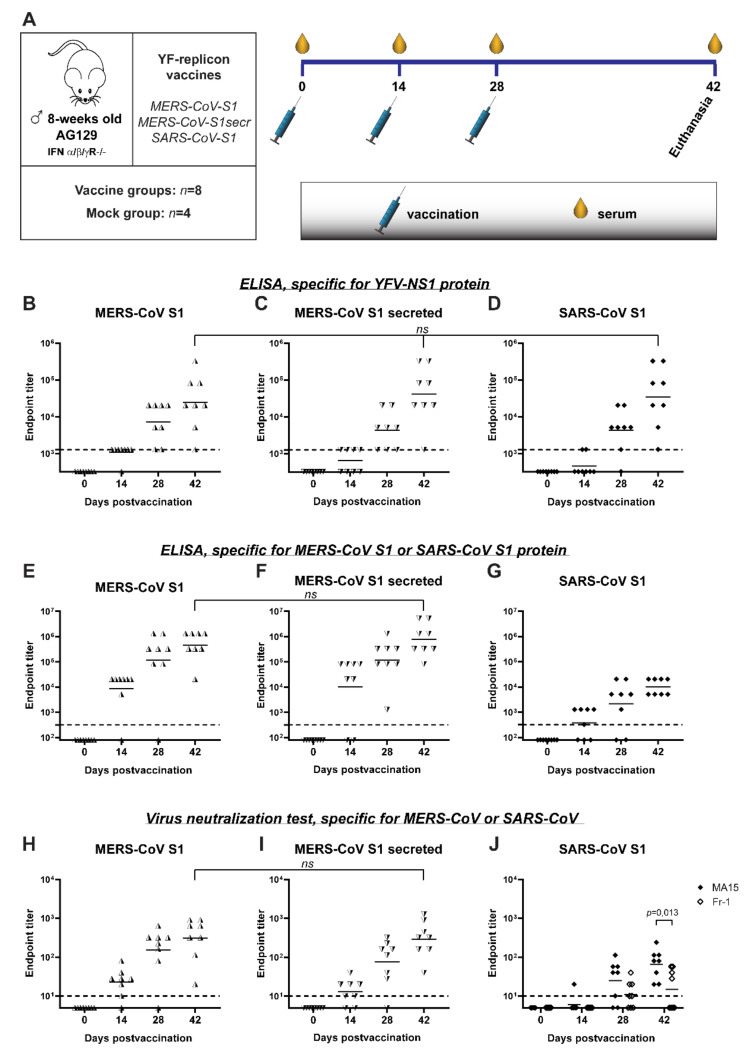
Humoral immune responses in mice, immunized with YF-replicons. (**A**) Schematic representation of the experimental design. (**B**–**D**) YFV-NS1-specific, (**E**,**F**) MERS-CoV S1-specific and (**G**) SARS-CoV S1-specific IgG antibodies, as measured by ELISA and expressed as endpoint titers. (**H**–**J**) Neutralizing titers against MERS-CoV (**H**,**I**) and against mouse-adapted SARS-CoV strain MA15 and a human-derived SARS-CoV strain Fr-1 (**J**) as measured by virus neutralization test and expressed as endpoint titers. The dotted lines indicate the lowest detectable titer (detection limit of the assay). All symbols represent individual titer values, and the geometric means are shown with horizontal lines. The YF-replicon used as a vaccine is shown above each graph. Statistical analysis was performed with nonparametric ANOVA (**B**–**D**) and with a *t*-test (**E**,**F**,**H**–**J**). Differences with *p* values ≤ 0.05 were considered significant and are shown where relevant. On the x-axis, days post-prime vaccination are shown.

**Table 1 vaccines-09-01492-t001:** Foreign protein inserts, derived from the spike proteins of MERS-CoV, SARS-CoV and MHV, inserted in the YF-replicons.

Foreign Insert (Domain *)	Virus	Strain	Amino Acids Used for the Inserts **
RBD	MERS-CoV	EMC2012	364–606
S1			18–750
Full length spike			18–1353
RBD	SARS-CoV	MA15	306–527
S1			14–665
RBD	MHV	A59	15–296
S1			15–712

* In the spike protein of the respective virus. ** Numbering in the respective spike proteins.

## Data Availability

All data was presented in the article, and this is NA.

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
