# Peer review of "A Yellow Fever 17D Virus Replicon-Based Vaccine Platform for Emerging Coronaviruses"

_vaccines, 2021, doi:10.3390/vaccines9121492_

Round 1
Reviewer 1 Report
This manuscript reports using a yellow fever 17D replicon-based vaccine platform for three coronaviruses (MERS, SARS and MHV). The molecular biology is excellent and the data are interesting and show that the vaccine is immunogenic, but unfortunately there are no challenge data to show the vaccines induce protective immunity. This point should be included in the Discussion.
A few points need attention. The authors should demonstrate that the correct folding of the Spike protein in the vaccine such that neutralizing epitopes are expressed. The easiest way of doing this would be to do western blots with monoclonal antibodies that recognize neutralizing epitopes.
The use of the term “SARS-CoV” is confusing as we have SARS-CoV-1 and SARS-CoV-2. Please use SARS-CoV-1.
Figure 1. I am confused why some panels appear to have more cells that others and the magnification of the panels appears different for some panels. Can the authors state in the Figure legend the magnification used for the panels in Figure 1? If the magnifications are different between panels they should be standardized to one magnification for all panels.
The mouse studies need some clarifications. I am unclear why the authors used AG129 mice as these are not immunocompetent and there are no challenge studies reported (which would not work for coronaviruses). I realize that 17D is lethal in AG129 mice but the “safety” section in the Discussion (line 680) is poor. There is no mention of body weights or clinical signs, no analysis of potential pathology in the mice. The authors should also explain why numbers of mice in each group differ in ELISA or neutralization positive. ELISA negative mice is not encouraging. Why not compare different doses of vaccine?
Lines 41-43 are very US/EU centric. 20 vaccines are now approved around the world and to focus on 3 approved in the US and 4 approved in EU does not do justice to what has been achieved.
Line 78: 17D
Line 87: The statement on Sanofi Pasteur and patents is not correct; they acquired the patents. The patenting of the 17D chimeras has a complicated history and I would suggest that this section is deleted.
Line 89: Similar “candidate” vaccines based. WN and Zika are not licensed.
Line 390: Please replace “creation” with “generation”.
Author Response
Open Review
We thank the reviewer for their questions and suggestions. Comments and answers are provided point-by-point below in blue. Line numbers mentioned refer to the (annotated) revised manuscript.
(x) I would not like to sign my review report
( ) I would like to sign my review report
English language and style
( ) Extensive editing of English language and style required
( ) Moderate English changes required
(x) English language and style are fine/minor spell check required
( ) I don't feel qualified to judge about the English language and style
Yes |
Can be improved |
Must be improved |
Not applicable |
|
Does the introduction provide sufficient background and include all relevant references? |
(x) |
( ) |
( ) |
( ) |
Is the research design appropriate? |
(x) |
( ) |
( ) |
( ) |
Are the methods adequately described? |
(x) |
( ) |
( ) |
( ) |
Are the results clearly presented? |
(x) |
( ) |
( ) |
( ) |
Are the conclusions supported by the results? |
(x) |
( ) |
( ) |
( ) |
Comments and Suggestions for Authors
This manuscript reports using a yellow fever 17D replicon-based vaccine platform for three coronaviruses (MERS, SARS and MHV). The molecular biology is excellent and the data are interesting and show that the vaccine is immunogenic, but unfortunately there are no challenge data to show the vaccines induce protective immunity. This point should be included in the Discussion.
Good animal models that support both replication of YF-17D and infections with MERS/SARS virus are not available at present, which complicates testing of protective efficacy. Establishing of a proper model susceptible for both YFV and MERS-CoV or SARS-CoV are therefore beyond the scope of this paper and will be tackled in follow-up experiments.
A few points need attention. The authors should demonstrate that the correct folding of the Spike protein in the vaccine such that neutralizing epitopes are expressed. The easiest way of doing this would be to do western blots with monoclonal antibodies that recognize neutralizing epitopes.
Western blots are blotted PAGE gels, and for this technique proteins are denatured with SDS, and are therefore not indicative for the correct folding of the protein of interest. Nonetheless, we do agree with the reviewer that it would be interesting to verify the correct folding of the domains expressed by our vector system. We think however, that the most important feature of the expressed proteins/domains is their immunogenicity. The fact that we detected neutralizing antibodies in sera of vaccinated mice is strong evidence that neutralizing epitopes are indeed exposed to the immune system and serve as an effective target for developing the antibodies. Also the correct glycosylation was confirmed, which suggests proper folding of the protein domains as well as correct membrane topology.
We included this point in the discussion (one before last paragraph, Modification M1).
The use of the term “SARS-CoV” is confusing as we have SARS-CoV-1 and SARS-CoV-2. Please use SARS-CoV-1.
The names according to the official ICTV nomenclature are SARS-CoV and SARS-CoV-2 (Coronaviridae Study Group of the International Committee on Taxonomy of Viruses, 2020 Nature Microbiology 5(4):536-544), while SARS-CoV-1 is NOT an official name for the virus that broke out in China in 2002. While we do understand that the names could be confusing, we do not feel free to use other names than the officially accepted.
Figure 1. I am confused why some panels appear to have more cells that others and the magnification of the panels appears different for some panels. Can the authors state in the Figure legend the magnification used for the panels in Figure 1? If the magnifications are different between panels they should be standardized to one magnification for all panels.
All images were made with the same magnification (x63 under immersion). When making the figure, we have scaled the images according to the bar that is printed on the images. When looking under the microscope, some cells are just bigger than others, which is also true for the nuclei. Also, cells in which the cytosol is stained look very bright and therefore may seem bigger than those showing surface expression.
We have added sentences for the magnification in Fig. 2 and Fig. 5 (these are the two figures that have microscopic images).
The mouse studies need some clarifications. I am unclear why the authors used AG129 mice as these are not immunocompetent and there are no challenge studies reported (which would not work for coronaviruses).
YFV and hence YF17D have a narrow host range, naturally infecting (except their mosquito vectors) only humans and non-human primates (G. K. Strode (Ed.): Yellow Fever: McGraw-Hill; First Edition; 1951). Likewise, in particular wild-type mice show a high level of resistance against YFV and low response to live YF17D immunization. This natural resistance can experimentally be overcome by genetic ablation of type I and II interferon signaling known to facilitate replication of many human RNA viruses in mice (PMID: 8009221; PMID: 7609046; PMID: 8825279). AG129 are type I and II interferon receptor deficient mice in the 129/env background, and are therefore highly permissive to YF17D and hence widely used for infection studies, as well as assessment of the safety and efficacy of YF17D-based vaccines (PMID: 31797751; PMID: 33260195; PMID: 19816561; PMID: 22425792; PMID: 32116148; PMID: 32265332; PMID: 30564463).
The abovementioned has already been partially addressed in the original Discussion (current lines 765 – 771: Immunocompetent mice, however, are inherently resistant to YF17D infection [59,91], and therefore require a booster vaccination regimen, ideally before anti-vector antibodies become detectable. In this regard, it was plausible that the replicon-based vaccines as investigated here, which do not spread within the host, would be even less potent in induction of antibody responses in mice. We therefore opted for using the AG129 mice, which are known to be very susceptible to infection with YF17D).
In order to provide more clarity, we also extended the part in the Results section, where the model is being introduced (please see M6 in the result section “S protein-expressing YF-replicons induce neutralizing antibodies against both MERS-CoV and SARS-CoV in mice”)
I realize that 17D is lethal in AG129 mice but the “safety” section in the Discussion (line 680) is poor. There is no mention of body weights or clinical signs, no analysis of potential pathology in the mice.
Current lines 655-657 (last paragraph of the result section) – a text had already been included in the Result section, stating that no adverse effect of the vaccine was observed in the mice in terms of clinical health and body weight. For this reason, no additional investigation has been carried out to check for pathology.
The authors should also explain why numbers of mice in each group differ in ELISA or neutralization positive. ELISA negative mice is not encouraging. Why not compare different doses of vaccine?
Considerable spread in the development of ELISA values and neutralising antibody titers among mice from the same experimental group is quite expected to our opinion, based on the general animal experimentation practice analysing immune responses. Most importantly however, we see significant general increase in the responses over time, and all animals show detectable (neutralising) antibody responses at day 42 after prime vaccination confirming that the vaccine is immunogenic and produces effective response for the antigen of interest in all mice.
Overall, the responses against the YFV-NS1 protein developed at later time points as compared to antibodies against S1 (both for MERS-CoV and SARS-CoV). This discrepancy could result from differences in immunogenicity and/or expression levels of NS1 and S1 respectively, surface exposure (we designed the S1 to be surface expressed, while NS1 is not expressed on the cell surface), or simply differences in the ELISA sensitivity, for example due to possible different affinities of the antibodies used.
Part of the text above has been included in the discussion (please see M2).
Why not compare different doses of vaccine?
In the described initial study, we tested a high vaccine dose to test for efficacy. In subsequent studies, dose-down regimes will be explored.
Lines 41-43 are very US/EU centric. 20 vaccines are now approved around the world and to focus on 3 approved in the US and 4 approved in EU does not do justice to what has been achieved.
Thank you for this remark, the introduction was modified accordingly (please see M3).
Line 78: 17D
Modified.
Line 87: The statement on Sanofi Pasteur and patents is not correct; they acquired the patents. The patenting of the 17D chimeras has a complicated history and I would suggest that this section is deleted.
We thank the reviewer for pointing this sensitive matter out. We think that the name ChimeriVax technology should be mentioned. However to avoid the complicated patent history, we removed the name of Sanofi Pasteur in the sentence about the pattents and added it later on as producer (please see line 95).
Line 89: Similar “candidate” vaccines based. WN and Zika are not licensed.
Thank you for this remark, we have modified the text (M4)
Line 390: Please replace “creation” with “generation”.
Modified.
Submission Date
01 September 2021
Date of this review
18 Sep 2021 16:05:21
Reviewer 2 Report
Summary: In this paper the authors describe their work optimizing a YF17D-vectored single-cycle replicon vaccine platform expressing the SARS-CoV and MERS-CoV spike proteins. The authors created single-cycle competent SARS-CoV or MERS-CoV surface protein pseudotyped virus particles using co-transfection of cells with RNA containing the antigen in the context of the YF17D genome, along with that containing the YF17D prM and E genes for packaging. Membrane-bound versions of the antigen which trafficked to the cell surface, which were visualized by microscopy. Furin sites were added to generate secreted forms of the antigen, which were analyzed by western blot and immunofluorescence. Mice were then inoculated 3x with the vaccine and serum levels of RBD, S1 and S protein binding Abs measured ELISA alongside virus neutralization using standard neutralization assays.
The manuscript is well-written and well-presented. However, only one vaccine immunogenicity study is provided with limited immunogenicity data, and this is done in a mouse strain more suited for safety testing than immunogenicity testing. This leads to difficulty in interpreting the results, particularly in the context the authors present them - as the validation of a vaccine platform they feel is likely to be excellent and comparable to the gold-standard YF17D live-attenuated vaccine. No such comparisons are made in the data, and the data presented suggest such excellent comparisons to be vastly premature.
Major Comments:
- Do the few remaining AAs of the YF-17D prM affect immunogenicity of the foreign antigen inserted? Do these change the antigen expressed?
- Is it possible to quantify the levels of antigen that make it to cell surface in Figure 2?
- The section title describing Fig 5 states that the replicons cause single-cycle infections, yet no data appears to be shown to confirm this. Please add or edit the language appropriately.
- Three doses of vaccine (two boosters) is impractical for public health in most parts of the world. Why were three doses, given 2 weeks apart, chosen rather than the more standard 2 doses, one month apart? Booster doses given too close together are known to generally be less efficacious.
- This paper would benefit greatly from more data on vaccine immunogenicity. Are T cell responses attained? This is particularly noteworthy as the discussion mentions the known T cell response data for YF17D vaccine responses (line 664). How would this compare?
- The assertion that this vaccine is better than RNA vaccines because YF17D elicits long-lasting, better protection is overstated, given the lack of data on how this construct actually performs relative to 17D. A single-cycle replicon is likely to induce lesser immune responses than a robust live-attenuated vaccine. No data comparisons between this platform and YF17D immunogenicity are made in this manuscript. For any statements to be made speculating that these vaccines induce the robust and long-lasted immunogenicity of YF17D, comparison with YF17D-vaccinated mice as a positive control would be necessary.
- While AG129 mice are justified for safety testing of the vectored vaccines described here relative to YF17D, they are inappropriate as a model for immunogenicity testing for replicating vaccines. The lack of type I/II IFN signaling may allow for significantly more robust replication relative to what would be seen in immunocompetent mice. At the same time, as mentioned in the discussion section, the success of YF17D is at least partly due to the ability of that vaccine to stimulate strong innate immune responses that drive robust adaptive immunity, so the lack of IFNa/b/g signaling capacity in this mouse model is a major barrier to true immunogenicity testing. Please justify, or most appropriately, run an additional study in Balb/c or another immunocompetent mouse model, testing with a 17D control, a single-cycle 17D replicon if possible, and adding T cell assays as well.
- Line 716 – the statement that immunocompetent mice need booster 17D vaccinations to induce antibody responses is untrue. While immunocompetent mice do not show signs of disease, high levels of antibody titers are induced in, for example, C57BL/6 mice after a single immunization (YF17D PRNTs ≥640). Again, this would be a key control to use as mentioned for comparison of this vaccine platform’s immunogenicity to the 17D vaccine.
- The vaccine seems to show slow seroconversion in the mice, 5/8 mice on d.14 isn’t a very solid percentage. The need for 3 doses of the SARS-CoV S1 vaccine to induce seroconversion of all animals is nonideal for a vaccine. Can the authors address implications?
- Line 673 – added safety of single-cycle replicons is excellent, but how does the immunogenicity compare to the live-attenuated virus? Would there be reason to believe that a multi-cycle 17D-vectored vaccine would have the same safety issues as YF17D, given that it would only include the non-structural proteins of 17D?
Minor Comments:
- The labeling of the X-axis in Figure 6 - I assume this refers to days post initial vaccination, not post last vaccination. This could be clarified in the legend.
- How does the dosage of 105 infectious/particles per animal compare to the 1 PFU mentioned at line 679?
- The authors should describe what inflamidase is, at line 707.
- The authors spend a lot of experimental energy confirming protein glycosylation in the paper, yet do not discuss glycosylation at all in the discussion. Perhaps some of this would be better presented as supplemental data, or its implications addressed better in the discussion.
- Line 192-193 – the authors state that the IVT reaction was conducted at 42C instead of the recommended 37C. No reason is given. Please explain, and possibly supply RNA characterization data as justification.
- Line 190 contains a typo “Invitrogen”
Author Response
We thank the reviewer for their questions and suggestions. Comments and answers are provided point-by-point below in blue. Line numbers mentioned refer to the (annotated) revised manuscript.
Open Review
( ) I would not like to sign my review report
(x) I would like to sign my review report
English language and style
( ) Extensive editing of English language and style required
( ) Moderate English changes required
(x) English language and style are fine/minor spell check required
( ) I don't feel qualified to judge about the English language and style
Yes |
Can be improved |
Must be improved |
Not applicable |
|
Does the introduction provide sufficient background and include all relevant references? |
(x) |
( ) |
( ) |
( ) |
Is the research design appropriate? |
( ) |
( ) |
(x) |
( ) |
Are the methods adequately described? |
(x) |
( ) |
( ) |
( ) |
Are the results clearly presented? |
(x) |
( ) |
( ) |
( ) |
Are the conclusions supported by the results? |
( ) |
(x) |
( ) |
( ) |
Comments and Suggestions for Authors
Summary: In this paper the authors describe their work optimizing a YF17D-vectored single-cycle replicon vaccine platform expressing the SARS-CoV and MERS-CoV spike proteins. The authors created single-cycle competent SARS-CoV or MERS-CoV surface protein pseudotyped virus particles using co-transfection of cells with RNA containing the antigen in the context of the YF17D genome, along with that containing the YF17D prM and E genes for packaging. Membrane-bound versions of the antigen which trafficked to the cell surface, which were visualized by microscopy. Furin sites were added to generate secreted forms of the antigen, which were analyzed by western blot and immunofluorescence. Mice were then inoculated 3x with the vaccine and serum levels of RBD, S1 and S protein binding Abs measured ELISA alongside virus neutralization using standard neutralization assays.
The summary of the reviewer indeed outlines the essence of the work described in the manuscript. However, we would like to point out that we do not claim that the replicon particles are pseudotyped with SARS-CoV or MERS-CoV surface proteins. The particles are generated using co-transfection of the replicon construct with an expression plasmid encoding the YFV glycoproteins. The produced virus particles express these original YF proteins on their surface, but not the MERS-CoV or SARS-CoV proteins. The latter is also confirmed by the results shown in Fig. 5 in the manuscript, which show that transfection with only the replicon construct encoding coronavirus spike domains does not result in the formation of infectious particles that can re-infect cells.
The manuscript is well-written and well-presented. However, only one vaccine immunogenicity study is provided with limited immunogenicity data, and this is done in a mouse strain more suited for safety testing than immunogenicity testing. This leads to difficulty in interpreting the results, particularly in the context the authors present them - as the validation of a vaccine platform they feel is likely to be excellent and comparable to the gold-standard YF17D live-attenuated vaccine. No such comparisons are made in the data, and the data presented suggest such excellent comparisons to be vastly premature.
AG129 mice are indeed used for safety testing. However, there are a number of reasons why we chose this model. First, as pointed out in the text, in contrast to humans, mice are intrinsically quite resistant to infection with YFV. There are only a couple of (natural) strains that allow a certain degree of replication, but those mice can likely only be used to test replication-competent vaccines. Our concern was that a non-spreading vector as we designed will be marginally efficient in immunocompetent mice, if at all, and because of this, its potential will severely be underestimated. AG129 mice are defective in IFN I response, but they have an intact adaptive immune system and do develop antibodies in response to antigenic stimulation. Second, as also pointed out in the text, neutralizing antibodies are a major immunogenic determinant in the protection against SARS-CoV and MERS-CoV and as recently demonstrated also against SARS-CoV-2. Therefore, in this first animal experiment, we were interested primarily in the capability of the replicon vector to elicit neutralizing antibodies. Third, “vaccination” of AG129 mice with YF17D is lethal. Therefore we were very interested in accumulating evidence that replicons are a safer option.
In the text we have added a paragraph to better explain our choice of the model, which has been widely used before to test both safety and efficacy of YF17D-based vaccines (see M6 in Results section near Fig.6)
Furthermore: we do not claim anywhere in the text that the replicon platform is excellent and comparable to the YF17D vaccine. We state that YF17D has “excellent safety and efficacy record” and that the replicons will benefit from this exceptional efficacy and immunogenicity of the back-bone used. No further comparisons or claims are made. To help modifying the impression of this reviewer about our claims, we toned down the conclusion paragraph in the discussion (last paragraph, please see M5).
Major Comments:
- Do the few remaining AAs of the YF-17D prM affect immunogenicity of the foreign antigen inserted? Do these change the antigen expressed?
Thank you for this interesting question, which to our opinion is rather difficult to answer with certainty without using methods such as X-ray crystallography, NMR spectroscopy and/or electron microscopy. We also suspect, that the effect of the remaining AAs may differ per insert.
However, since the foreign antigen is expressed well and shows trafficking and localization as expected, and is immunogenic (elicits neutralizing antibodies), it can be expected that the immunogens that we tested are very similar in structure compared to the naturally expressed proteins.
- Is it possible to quantify the levels of antigen that make it to cell surface in Figure 2?
Such analysis would likely be very laborious and probably require specialized equipment which is not available to us. Answering the question would probably be easier using very careful cell fractionation experiments instead of using these microscopic images. However, we do not really see the importance of quantifying the amount of protein that makes it to the cell surface, as it is impossible to know how the differently localized populations contribute to immunogenicity. Moreover, the trafficking of proteins is a continuous and dynamic process and measurements will always be a one time-point representation of the situation, which will therefore not really answer the question.
- The section title describing Fig 5 states that the replicons cause single-cycle infections, yet no data appears to be shown to confirm this. Please add or edit the language appropriately.
We understand that we do not directly show this in the figure, however indirectly it can be concluded from these results: only when both the replicon and the plasmid excoding prM/E are co-transfected, the resulting virus particles can enter cells and express the foreign antigen (panels on the right). If in that situation there would be a second (and subsequent) round possible then new particles would have to be made without the presence of the genetic information for expression of the prM/E surface proteins, since this information is not encoded on the replicon. If this were possible however, then transfection of the replicon alone would also have resulted in infectious particles that would then infect new cells and express the foreign proteins encoded. However, this was not the case (middle panels), confirming that the replicon particles only support a single cycle of replication and no new particles are possible.
In order to avoid confusion we have removed the words from the title but added a sentence of this paragraph in the Results to clarify (please see M9).
- Three doses of vaccine (two boosters) is impractical for public health in most parts of the world. Why were three doses, given 2 weeks apart, chosen rather than the more standard 2 doses, one month apart? Booster doses given too close together are known to generally be less efficacious.
The scope of this proof-of-concept study was to present the design, construction and preliminary immunogenicity of our new YF17D replicon platform, rather than in-depth assessment of vaccine pharmacodynamics or pharmacokinetics. The actual dosing schedule was hence based on our in-house experience for live-attenuated YF17D, with the aim to booster humoral immunity in the most efficient way in a short time, however not to mimick or imply any clinical immunization schedule. Furthermore, two booster doses were given to ensure sufficient exposure considering the fairly low titer of the vaccine available per dose.
To clarify these issues, we now added text in the M&M section, under section “Mouse experiment” (please see M7), and in the main discussion (please see M8)
- This paper would benefit greatly from more data on vaccine immunogenicity. Are T cell responses attained? This is particularly noteworthy as the discussion mentions the known T cell response data for YF17D vaccine responses (line 664). How would this compare?
We do agree with the reviewer that more details about immunogenicity will benefit the development of the vaccine platform. Our first focus was on antibodies, since, as already stated above, those are the major efficacy determinant for vaccines against human coronaviruses,
In future studies, we will address the cellular responses and Th1/Th2 signatures in more details.
- The assertion that this vaccine is better than RNA vaccines because YF17D elicits long-lasting, better protection is overstated, given the lack of data on how this construct actually performs relative to 17D. A single-cycle replicon is likely to induce lesser immune responses than a robust live-attenuated vaccine. No data comparisons between this platform and YF17D immunogenicity are made in this manuscript. For any statements to be made speculating that these vaccines induce the robust and long-lasted immunogenicity of YF17D, comparison with YF17D-vaccinated mice as a positive control would be necessary.
To our opinion there is no claim in the text at all that the vaccine developed with this work, nor the YF17D vaccine, are better than RNA vaccines, and this has never been our intention, so it is unclear to us how the reviewer has drawn this conclusion.
For the remaining part of this comment, please see our explanations given above.
- While AG129 mice are justified for safety testing of the vectored vaccines described here relative to YF17D, they are inappropriate as a model for immunogenicity testing for replicating vaccines. The lack of type I/II IFN signaling may allow for significantly more robust replication relative to what would be seen in immunocompetent mice. At the same time, as mentioned in the discussion section, the success of YF17D is at least partly due to the ability of that vaccine to stimulate strong innate immune responses that drive robust adaptive immunity, so the lack of IFNa/b/g signaling capacity in this mouse model is a major barrier to true immunogenicity testing. Please justify, or most appropriately, run an additional study in Balb/c or another immunocompetent mouse model, testing with a 17D control, a single-cycle 17D replicon if possible, and adding T cell assays as well.
YFV and hence YF17D have a narrow host range, naturally infecting (except their mosquito vectors) only humans and non-human primates (G. K. Strode (Ed.): Yellow Fever: McGraw-Hill; First Edition; 1951). Likewise, in particular wild-type mice show a high level of resistance against YFV and low response to live YF17D immunization. This natural resistance can experimentally be overcome by genetic ablation of type I and II interferon signaling known to facilitate replication of many human RNA viruses in mice (PMID: 8009221; PMID: 7609046; PMID: 8825279). AG129 are type I and II interferon receptor deficient mice in the 129/env background that are highly permissive to YF17D but they have an intact adaptive immune system and hence they are widely used for the assessment of both the safety and efficacy of YF17D-based vaccines. This has already been mentioned in the original Discussion (Lines referred to here : “Immunocompetent mice, however, are inherently resistant to YFV-17D infection [59,91], and therefore require a booster vaccination with an infection-competent virus, ideally before anti-vector antibodies become detectable. In this regard, it was plausible that the replicon-based vaccines as investigated here, which do not spread within the host, would be even less potent in induction of antibody responses in mice. We therefore opted for using the AG129 mice, which are known to be very susceptible to infection with YFV-17D.”)
We added a short narrative to the result section, where the animal model is being introduced, for further justification (please see M6).
- Line 716 – the statement that immunocompetent mice need booster 17D vaccinations to induce antibody responses is untrue. While immunocompetent mice do not show signs of disease, high levels of antibody titers are induced in, for example, C57BL/6 mice after a single immunization (YF17D PRNTs ≥640). Again, this would be a key control to use as mentioned for comparison of this vaccine platform’s immunogenicity to the 17D vaccine.
As stated above, YF17D is severely attenuated in immunocompetent mice, including C57BL/6 (Andrea K. Erickson and Julie K. Pfeiffer, doi: 10.1099/vir.0.000075). We were unable to find the work that the reviewer is referring to that would state “YF17D PRNTs ≥640”. Since the replicon platform generates particles that are uncapable of spread, those particles would be even more attenuated than YF17D. Therefore, finding the right mouse model in which to compare our platform and YF17D is rather complicated and perhaps a set of pilot experiments will be needed in advance. The point raised by the reviewer is definitely important and interesting to tackle, but with the outlined problematic situation, it is beyond the scope of the current work.
- The vaccine seems to show slow seroconversion in the mice, 5/8 mice on d.14 isn’t a very solid percentage. The need for 3 doses of the SARS-CoV S1 vaccine to induce seroconversion of all animals is nonideal for a vaccine. Can the authors address implications?
This indeed is a notion that is often used when selection for vaccine candidates needs to be made. However, we are currently witnessing discussions about “third” dose against SARS-CoV-2, which has already been implemented in some countries (for instance Israel) for their total population, while in others is advised for “high-risk” population groups. Furthermore, vaccines against Hepatitis A/B and against HPV also have a three-dose regimen. Therefore, it is obvious that the three-dose regimen is not always a “no-go”, but has to be put in context.
In addition, there are numerous studies in which not all vaccinated mice seroconvert 14 days after vaccination (this is the measurement that reflects the response after the prime vaccination).
We find it positive that already after two vaccinations, the majority of mice (21 out of 24) had detectable ELISA antibodies and 20 out of 24 had detectable neutralizing antibodies. These percentages go up to 100% after the third vaccination.
Moreover, the regime used in these mice can neither be compared, or be a good guideline for what would be needed in humans. It only shows proof-of-principle. Once more preclinical work has been done, the regime in humans has to be evaluated through phase 1/2 trials, and this will not necessarily be based on what has been used in these mice.
- Line 673 – added safety of single-cycle replicons is excellent, but how does the immunogenicity compare to the live-attenuated virus? Would there be reason to believe that a multi-cycle 17D-vectored vaccine would have the same safety issues as YF17D, given that it would only include the non-structural proteins of 17D?
The mechanism behind the severe adverse effects resulting from vaccination with YF17D are not well understood, because of their rarity and difficulty to study. It has been observed that in people with vaccine-associated viscerotropic disease, there is high-level, prolonged viremia, suggestive of inability to control virus replication and spread (Barrett and Teuwen, 2009, doi:10.1016/j.coi.2009.05.018). Multi-cycle 17D will therefore likely also be difficult to control in these people and likely cause similar adverse effects as YF17D itself. In that regard, replicons, even after multiple applications, would not be capable of causing viral dissemination, because they cannot spread. They would therefore not cause these problems.
Regarding the replicon efficacy and comparison with YF17D vaccine, this question has been asked already with different wording in the previous points and we have provided answers/comments (please see responses above).
Minor Comments:
- The labeling of the X-axis in Figure 6 - I assume this refers to days post initial vaccination, not post last vaccination. This could be clarified in the legend.
Thank you for this notion. We included a sentence in the legend of Fig. 6 to clarify this point.
- How does the dosage of 105 infectious/particles per animal compare to the 1 PFU mentioned at line 679?
To our opinion, non-spreading replicon particles cannot be compared with spreading-competent virus.
- The authors should describe what inflamidase is, at line 707.
We added a brief explanation, please see line 723-724.
- The authors spend a lot of experimental energy confirming protein glycosylation in the paper, yet do not discuss glycosylation at all in the discussion. Perhaps some of this would be better presented as supplemental data, or its implications addressed better in the discussion.
Glycosylation generally plays an important role in the immunogenicity of a protein. One of the main draw-backs of bacterial and yeast-based platforms that are used to produce glycoproteins is glycosylation patters that differ from those produced in mammalian cells, or they are absent. By showing that the expressed proteins are glycosylated as expected, adds to the claim that native-like proteins are produced from this platform. Furthermore, it confirms correct folding and membrane topology, since if those were incorrect, the proteins are likely to show aberrant glycosylation and end up on the wrong side of the membrane.
Please see the modified text in the manuscript marked as M1 in the discussion section (one before last paragraph) where we have added clarification on this point.
- Line 192-193 – the authors state that the IVT reaction was conducted at 42C instead of the recommended 37C. No reason is given. Please explain, and possibly supply RNA characterization data as justification.
The constructs that we have used contain an SP6 promotor for in vitro RNA production. SP6 polymerases work most efficiently at higher temperatures and although most manufacturers recommend 37oC, in our experience 42oC is more preferrable and that is why we standardly use this temperature.
- Line 190 contains a typo “Invitrogen”
Thank you for this comment, the mistake was corrected
Submission Date
01 September 2021
Date of this review
27 Sep 2021 08:14:59
Round 2
Reviewer 2 Report
The responses to comments 1-5 and 9-10 are acceptable, as well as the responses to the minor comments. The responses to comments 6, 7 and 8 are all related and display a misunderstanding of mouse models of yellow fever vaccination versus infection.
____________
The main objection I have to this vaccine study is still that the mouse model system used is inappropriate, and the authors have defended its use inappropriately. While the current studies are informative to some degree, use of AG129 mice is insufficient as the only model system used for the entirety of this study. The other reviewer also had the same question about the mouse strain used. Sole use of AG129 mice is not appropriate for immunogenicity studies of a coronavirus vaccine, regardless of the vaccine backbone. This statement made by the authors in response to this criticism, in fact, highlights the issue with using this mouse strain for immunogenicity testing:
“AG129 mice are indeed used for safety testing. However, there are a number of reasons why we chose this model.”… “Our concern was that a non-spreading vector as we designed will be marginally efficient in immunocompetent mice, if at all”
Is this not a key problem with these studies?
The authors assert that this mouse strain is both necessary and appropriate for studies of YF17D-vectored vaccines, and that YF17D cannot be used in wild-type mice. This is untrue. The authors cite a number of articles as evidence. None of those come to this conclusion. Approximately half of these studies solely discuss lethal mouse models of YFV infection, rather than mouse models of yellow fever vaccination. YF17D does not disseminate in wild-type mice – not surprising, considering this is an attenuated vaccine – but it certainly induces robust immunity, allowing wild-type mouse models to be perfectly usable for 17D-vectored vaccines.
The other half of the mentioned studies report on 17D-vectored vaccines – and while these often use AG129 mice for some studies, typically of vaccine efficacy – most of these also appropriately demonstrate immunogenicity in immunocompetent BL/6 or Balb/c mice, acknowledging the major issue that immunocompromised mice do not provide accurate representations of the magnitude of immune responses induced by replicating vaccines. In fact, Figure 5 of one of the cited papers (PMID 30564463), clearly illustrates this problem, demonstrating the impact of an intact IFNAR response on Ab titers in mice (massive Ab titers to their 17D-vectored Zika vaccine are significantly reduced, though still present, in immunocompetent mice).
Other coronavirus vaccines induce significant neutralizing antibodies in immunocompetent mice, and these are the data that would be necessary to meaningfully cross-compare with other such vaccines. As another alternative, hamsters are an appropriate immunocompetent rodent model for both coronaviruses and yellow fever.
"Likewise, in particular wild-type mice show a high level of resistance against YFV and low response to live YF17D immunization."
The first part of this sentence is true; the second half is untrue.
"Immunocompetent mice, however, are inherently resistant to YF17D infection [59,91], and therefore require a booster vaccination regimen, ideally before anti-vector antibodies become detectable."
Similarly, this is inaccurate - where does this come from? Obtain a dose of the YF17DD vaccine, reconstitute it, and deliver 10^4 PFU into a mouse’s footpad – then measure YF neutralizing antibodies and T cells 14 days later and this is readily proved. The cited papers are referring to dissemination of YF17D in mice, not YF17D’s ability to induce robust immune responses in mice (which they can and do, despite lack of viral dissemination). See PMID 25539816 and 12036323 for illustrations of antibody and T cell responses, respectively.
A final small study of immunogenicity of these vaccines in immunocompetent mice or hamsters, as others have provided for other 17D-vectored vaccines, would address these issues and should be done.
______________
Lines 816-818: "It is justified to expect that the replicons will (at least in part) benefit from the exceptional efficacy and immunogenicity of the YF17D backbone, but that remains to be determined in detail in future studies."
This is still vastly overstated. Why is it justified? Any citations comparing the exceptional efficacy and longevity of the YF17D response to those of 17D-vectored vaccines (and in this case, a one-cycle 17D vectored vaccine) are absent and must be added for this statement to be made, otherwise it must be edited further.
Author Response
Answers and comments to this reviewer are provided below in blue.
The responses to comments 1-5 and 9-10 are acceptable, as well as the responses to the minor comments. The responses to comments 6, 7 and 8 are all related and display a misunderstanding of mouse models of yellow fever vaccination versus infection.
We thank the reviewer for acknowledging that our responses to most comments are acceptable. Indeed, the only issue left concerns the use of AG129 mice for our proof-of-principle (immunogenicity) experiment showing neutralizing antibody response towards the viruses of interest using our vaccine design. We respond to the reviewer’s arguments as follows below.
The main objection I have to this vaccine study is still that the mouse model system used is inappropriate, and the authors have defended its use inappropriately. While the current studies are informative to some degree, use of AG129 mice is insufficient as the only model system used for the entirety of this study.
We fully agree that a study in AG129 mice (alike in any single mouse model) will not suffice to validate our vaccine platform as presented in this paper for development into clinical studies. Due to the obvious host-range restrictions of yellow fever virus (and flaviviruses in general), no single mouse model does fulfills all criteria for any YF17D-based vaccine to proceed to clinical studies. However, use of IFNR-KO mice allows at least to follow primary pharmacodynamics (i.e. adaptive immune responses), and there is no doubt that AG129 mice fulfill the latter requirement. The entire flavivirus field uses this mouse strain, including for immunological mechanistic studies.
Our first study emphasizes at validating the overall vaccine design, with a focus on the correct expression, intracellular localization and targeting of the antigens of interest. We hence decided to limit our proof-of-principle to a first immunogenicity assessment in a small animal model that supports sufficient replication of the vaccine vector and hence expression of its protein products, respectively. With this respect any susceptible mouse model is to be considered valid. Future studies will include challenge studies in small animals (Syrian golden hamsters) and non-human primates to further substantiate the data, but this is beyond the scope of the current study. The reviewer in his/her phrase above does acknowledge the value of the study as performed. As the model we employed is capable of developing a (neutralizing) antibody response upon vaccination with our designed vaccine candidates, we do think it is well-suited as a model for this proof-of-principle experiment.
In any case, we wanted to avoid any overclaiming of results by our serology assessment. Of note, the magnitude of humoral responses seen in Type I and II IFN deficient mice can rather be expected at the lower range compared to other mouse models (van den Broek et al., Immunol Rev. 1995. PMID: 8825279; Fink et al., Eur J Immunol. 2006. PMID: 16810635). We are hence convinced that with the choice of AG129 we avoid generating and reporting irrelevant or exaggerated levels of nAbs.
The other reviewer also had the same question about the mouse strain used. Sole use of AG129 mice is not appropriate for immunogenicity studies of a coronavirus vaccine, regardless of the vaccine backbone.
It remains a little unclear what the exact arguments for this statement are, other than that the reviewer states that AG129 have predominantly been used for safety testing by others. Although these mice are partially deficient in their innate immune system, their adaptive system is intact and they display robust antibody response upon immunization, as we could show in our study using our vaccine candidates. The competence of the model has recently been confirmed again by Mishra, Kum and co-workers in several publications: NPJ Vaccines (PMID: 30564463), mBio (PMID: 32265332) and EMI (PMID: 32116148), which consistently show that AG129 mice display both vigorous T-cell and protective antibody (nAb and Fc-mediated activity) responses upon immunization with a YF17D based vaccine.
This statement made by the authors in response to this criticism, in fact, highlights the issue with using this mouse strain for immunogenicity testing:
“AG129 mice are indeed used for safety testing. However, there are a number of reasons why we chose this model.”… “Our concern was that a non-spreading vector as we designed will be marginally efficient in immunocompetent mice, if at all”
Is this not a key problem with these studies?
We respectfully disagree with the reviewer on this point. Small animal models will always be suboptimal compared to the natural host, in this case humans, for which the vaccine is meant. We therefore think that a model such as AG129, in which the levels of replication of the YF17D vector mimics that in humans well, is not less valuable than an immunocompetent mouse model in which replication and expression is marginal, so incomparable to humans. While the AG129 model is partially deficient in its innate immune response, it is well-capable of eliciting (neutralizing) antibody response upon immunization, and that is what we were looking for as initial proof-of-principle in our experiment.
The authors assert that this mouse strain is both necessary and appropriate for studies of YF17D-vectored vaccines, and that YF17D cannot be used in wild-type mice. This is untrue. The authors cite a number of articles as evidence. None of those come to this conclusion. Approximately half of these studies solely discuss lethal mouse models of YFV infection, rather than mouse models of yellow fever vaccination. YF17D does not disseminate in wild-type mice – not surprising, considering this is an attenuated vaccine – but it certainly induces robust immunity, allowing wild-type mouse models to be perfectly usable for 17D-vectored vaccines.
We have not stated that the mouse strain used is “necessary”, but we do think it is appropriate for the study of YF17D-based vaccines, as argued above. In the meantime, the group of Dr. Dallmeier has conducted a detailed study comparing five different mouse models, including wildtype BALB/c and C57BL/6 (B6) mice (Ji Ma et al. accepted for publication pending minor revision).Here they measured nAb, total IgG, CMI and Th polarization and kinetics of YF17D-specific immune responses following vaccination with a dose-range from 2 to 2x104 PFU of the commercial YF17D vaccine, followed by intracranial challenge with a 100x LD50 dose. This study clearly demonstrates that a lot of textbook knowledge regarding the mechanism of action of YF17D that was derived from mouse studies is to be read (or re-written) carefully. In a nutshell, at high doses 17D provides full protection against massive lethal YF challenge in any mouse model. At low doses, the picture is more differentiated, and likely more relevant. E.g. 100% of BALB/c mice vaccinated with as little as 2-20 PFU of 17D survive challenge despite very low IgG and no measurable nAb. In contrast, B6 mice succumb under the same conditions. Likewise, IFNAR-/- have extremely high nAb, IgG and CMI levels, and survive despite they cannot eliminate YF17D from the brain. In the latter model, inactivated 17D had no activity nor survival benefit whatsoever anymore, clearly demonstrating that the replication competence of the vector in vivo is key to the efficacy of 17D and 17D-derived vaccines. We hence honestly disagree with the very generic statement that WT mice should be in general well suitable for assessment of YF17D immunity. Historical reports which may serve here as benchmark tend to overdose the vaccine to force a response (to overcome host-restriction).
The other half of the mentioned studies report on 17D-vectored vaccines – and while these often use AG129 mice for some studies, typically of vaccine efficacy – most of these also appropriately demonstrate immunogenicity in immunocompetent BL/6 or Balb/c mice, acknowledging the major issue that immunocompromised mice do not provide accurate representations of the magnitude of immune responses induced by replicating vaccines. In fact, Figure 5 of one of the cited papers (PMID 30564463), clearly illustrates this problem, demonstrating the impact of an intact IFNAR response on Ab titers in mice (massive Ab titers to their 17D-vectored Zika vaccine are significantly reduced, though still present, in immunocompetent mice).
Indeed our work by Kum et al, 2018 (PMID: 30564463), as referred to by the reviewer, we clearly demonstrate Ab responses in immunocompetent mice, yet clearly and significantly lower than in IFNAR-/- mice. However the question is whether this is because of an aberrant immune response in the IFNAR-/- mice, resulting in high antibody responses as the reviewer suggests, or is it because of very low-level replication of the virus/vaccine in the immunocompetent mice, which results in lower immune response including lower antibody levels? And is this immunocompetent rodent model then representative for a situation in humans where virus replication and expression is more robust?
We as authors of the mentioned paper showed an additional experiment, where the question of aberrant antibody response is addressed by blocking the IFN receptor in WT mice. We show a significant increase in antibody responses in C57BL/6 mice, but not in BALB/c mice and conclude that the response depends largely on the WT mouse strain used (see also response above).
To our opinion the Reviewer’s arguments actually support our view that IFN deficient mice are suitable models for analyzing immunogenicity after 17D vaccination, while WT mice may have serious limitations.
Other coronavirus vaccines induce significant neutralizing antibodies in immunocompetent mice, and these are the data that would be necessary to meaningfully cross-compare with other such vaccines. As another alternative, hamsters are an appropriate immunocompetent rodent model for both coronaviruses and yellow fever.
"Likewise, in particular wild-type mice show a high level of resistance against YFV and low response to live YF17D immunization."
The first part of this sentence is true; the second half is untrue.
"Immunocompetent mice, however, are inherently resistant to YF17D infection [59,91], and therefore require a booster vaccination regimen, ideally before anti-vector antibodies become detectable."
Similarly, this is inaccurate - where does this come from? Obtain a dose of the YF17DD vaccine, reconstitute it, and deliver 10^4 PFU into a mouse’s footpad – then measure YF neutralizing antibodies and T cells 14 days later and this is readily proved. The cited papers are referring to dissemination of YF17D in mice, not YF17D’s ability to induce robust immune responses in mice (which they can and do, despite lack of viral dissemination). See PMID 25539816 and 12036323 for illustrations of antibody and T cell responses, respectively.
This paper shows T-cell immune response in C57BL/6 (wt B6) mice (neutralizing antibodies and T-cell responses)
We thank the reviewer for pointing out the papers PMID 25539816 and 12036323. In those works, indeed neutralizing responses against YFV-17D and T-cell responses against YFV proteins were clearly measurable in wt (C57BL/6(J)) mice and the first manuscript also shows protection in mice against otherwise lethal infection with YFV17D via the i.c. route. We have further revised the manuscript text to accommodate this information (lines 727 – 731 in the revised manuscript). In the same text passage, we have modified the statement “"Immunocompetent mice, however, are inherently resistant to YF17D infection” to “In immunocompetent mice, however, YF17D is severely attenuated [59,91]” to better describe the findings in the two cited works. We further refer to our argumentation and our study by Ma Ji et al. cited above. If required and the handling Journal Editor allows, we can provide the last revised version to the reviewer.
We have also added the word “very” in line 588 to underline the permissiveness of the AG129 model to infection with YF17D.
A final small study of immunogenicity of these vaccines in immunocompetent mice or hamsters, as others have provided for other 17D-vectored vaccines, would address these issues and should be done.
As explained above, we have already performed a small immunogenicity study and we have shown immunogenicity of the vaccine in the AG129 mice. We also explained why, to our opinion, the AG129 model is relevant in this proof-of-principle study. The detected antibody responses to our opinion indicate that the antigens expressed from our vaccine are immunogenic, which is what we wanted to confirm after testing and validating the molecular aspects of the vaccine design.
_____________
Lines 816-818: "It is justified to expect that the replicons will (at least in part) benefit from the exceptional efficacy and immunogenicity of the YF17D backbone, but that remains to be determined in detail in future studies."
This is still vastly overstated. Why is it justified? Any citations comparing the exceptional efficacy and longevity of the YF17D response to those of 17D-vectored vaccines (and in this case, a one-cycle 17D vectored vaccine) are absent and must be added for this statement to be made, otherwise it must be edited further.
We think it is justified since the YF17D backbone forms a large part of the vaccine, and we therefore expect it to behave in a similar way as the original YF17D vaccine. We have changed the wording as to indicate that this is OUR expectation, but still clearly state that this should be experimentally confirmed. We further add a reference to a review article in Nature Reviews in Microbiology (Draper SJ, Heeney JL. Viruses as vaccine vectors for infectious diseases and cancer. Nat Rev Microbiol. 2010 Jan;8(1):62-73. doi: 10.1038/nrmicro2240. PMID: 19966816.) sharing our opinion, based on which this statement was also inspired. In this way this may be regarded a strong opinion, yet not really an overstatement. Please see the modification in the text, line 821.
Moreover, in our own study by Sanchez-Felipe et al. published in Nature (PMID: 33260195) we clearly show that the humoral and cellular immunity elicited against the target antigen that is vectored by 17D benefits from the vector. In particular the CD4+ and CD8+ responses seen appear stronger than those reported for many other COVID vaccines and vaccine candidates.
Our statement on the benefit of using 17D as a vector is further supported by our study on a YF17D-vectored HBV core (HBc) vaccine. Here we quantitatively and qualitatively compared the immune profile elicited in mice against HBc: head-to-head for (i) recombinant HBc protein, (ii) HBc vectored by an Adenovirus and (iii) HBc vectored by 17D (Boudewijns R, Ma J, Neyts J, Dallmeier K. A novel therapeutic HBV vaccine candidate induces strong polyfunctional cytotoxic T cell responses in mice. JHEP Rep. 2021 Apr 22;3(4):100295. doi: 10.1016/j.jhepr.2021.100295. PMID: 34159304; PMCID: PMC8203848.). In particular, regarding the particularly strong Th and CTL responses YF17D outcompeted any other platform tested.